# Revisiting Hilbert-Schmidt Information Bottleneck for Adversarial Robustness

**Zifeng Wang**[*]
Northeastern University
zifengwang@ece.neu.edu

**Tong Jian**[*]
Northeastern University
jian@ece.neu.edu

**Aria Masoomi**
Northeastern University
masoomi.a@northeastern.edu

**Stratis Ioannidis**
Northeastern University
ioannidis@ece.neu.edu

**Jennifer Dy**
Northeastern University
jdy@ece.neu.edu

a

## Abstract

We investigate the HSIC (Hilbert-Schmidt independence criterion) bottleneck as a regularizer for learning an adversarially robust deep neural network classifier. In addition to the usual cross-entropy loss, we add regularization terms for every intermediate layer to ensure that the latent representations retain useful information for output prediction while reducing redundant information. We show that the HSIC bottleneck enhances robustness to adversarial attacks both theoretically and experimentally. In particular, we prove that the HSIC bottleneck regularizer reduces the sensitivity of the classifier to adversarial examples. Our experiments on multiple benchmark datasets and architectures demonstrate that incorporating an HSIC bottleneck regularizer attains competitive natural accuracy and improves adversarial robustness, both with and without adversarial examples during training. Our code and adversarially robust models are publicly available.[2]

## 1 Introduction

Adversarial attacks [8, 17, 18, 3, 5] to deep neural networks (DNNs) have received considerable attention recently. Such attacks are intentionally crafted to change prediction outcomes, e.g, by adding visually imperceptible perturbations to the original, natural examples [25]. Adversarial robustness, i.e., the ability of a trained model to maintain its predictive power under such attacks, is an important property for many safety-critical applications [4, 6, 26]. The most common approach to construct adversarially robust models is via adversarial training [34, 36, 30], i.e., training the model over adversarially constructed samples.

Alemi et al. [1] propose using the so-called *Information Bottleneck* (IB) [27, 28] to ehnance adversarial robustness. Proposed by Tishby and Zaslavsky [28], the information bottleneck expresses a tradeoff between (a) the mutual information of the input and latent layers vs. (b) the mutual information between latent layers and the output. Alemi et al. show empirically that using IB as a learning objective for DNNs indeed leads to better adversarial robustness. Intuitively, the IB objective increases the entropy between input and latent layers; in turn, this also increases the model's robustness, as it makes latent layers less sensitive to input perturbations.

Nevertheless, mutual information is notoriously expensive to compute. The Hilbert-Schmidt independence criterion (HSIC) has been used as a tractable, efficient substitute in a variety of machine

---

[*]Equal contribution.
[2]https://github.com/neu-spiral/HBaR

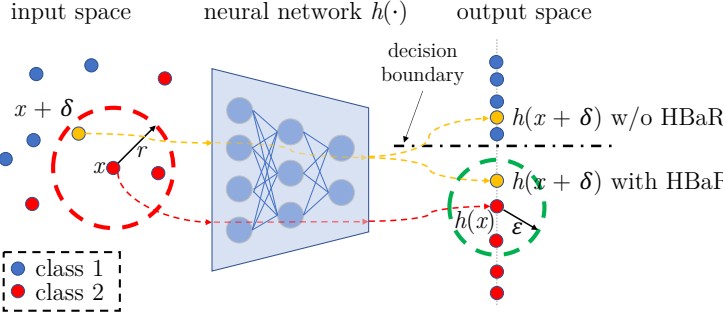

Figure 1: Illustration of HBaR for adversarial robustness. A neural network trained with HBaR gives a more constrained prediction w.r.t. perturbed inputs. Thus, it is less sensitive to adversarial examples.

learning tasks [31, 32, 33]. Recently, Ma et al. [16] also exploited this relationship to propose an *HSIC bottleneck* (HB), as a variant to the more classic (mutual-information based) information bottleneck, though not in the context of adversarial robustness.

We revisit the HSIC bottleneck, studying its adversarial robustness properties. In contrast to both Alemi et al. [1] and Ma et al. [16], we use the HSIC bottleneck as a regularizer in addition to commonly used losses for DNNs (e.g., cross-entropy). Our proposed approach, HSIC-Bottleneck-as-Regularizer (HBaR) can be used in conjunction with adversarial examples; even without adversarial training, it is able to improve a classifier's robustness. It also significantly outperforms previous IB-based methods for robustness, as well as the method proposed by Ma et al.

Overall, we make the following contributions:

1. We apply the HSIC bottleneck as a regularizer for the purpose of adversarial robustness.
2. We provide a theoretical motivation for the constituent terms of the HBaR penalty, proving that it indeed constrains the output perturbation produced by adversarial attacks.
3. We show that HBaR can be naturally combined with a broad array of state of the art adversarial training methods, consistently improving their robustness.
4. We empirically show that this phenomenon persists even for weaker methods. In particular, HBaR can even enhance the adversarial robustness of plain SGD, without access to adversarial examples.

The remainder of this paper is structured as follows. We review related work in Sec. 2. In Sec. 3, we discuss the standard setting of adversarial robustness and HSIC. In Sec. 4, we provide a theoretical justification that HBaR reduces the sensitivity of the classifier to adversarial examples. Sec. 5 includes our experiments; we conclude in Sec. 6.

## 2 Related Work

**Adversarial Attacks.** Adversarial attacks often add a constrained perturbation to natural inputs with the goal of maximizing classification loss. Szegedy et al. [25] learn a perturbation via box-constrained L-BFGS that misleads the classifier but minimally distort the input. FGSM, proposed by Goodfellow et al. [8], is a one step adversarial attack perturbing the input based on the sign of the gradient of the loss. PGD [13, 17] generates adversarial examples through multi-step projected gradient descent optimization. DeepFool [18] is an iterative attack strategy, which perturbs the input towards the direction of the decision boundaries. CW [3] applies a rectifier function regularizer to generate adversarial examples near the original input. AutoAttack (AA) [5] is an ensemble of parameter-free attacks, that also deals with common issues like gradient masking [19] and fixed step sizes [17].

**Adversarial Robustness.** A common approach to obtaining robust models is *adversarial training*, i.e., training models over adversarial examples generated via the aforementioned attacks. For example, Madry et al. [17] show that training with adversarial examples generated by PGD achieves good robustness under different attacks. DeepDefense [34] penalizes the norm of adversarial perturbations. TRADES [36] minimizes the difference between the predictions of natural and adversarial

examples to get a smooth decision boundary. MART [30] pays more attention to adversarial examples from misclassified natural examples and adds a KL-divergence term between natural and adversarial samples to the cross-entropy loss. *We show that our proposed method HBaR can be combined with several such state-of-the-art defense methods and boost their performance.*

**Information Bottleneck.** The information bottleneck (IB) [27, 28] expresses a tradeoff in latent representations between information useful for output prediction and information retained about the input. IB has been employed to explore the training dynamics in deep learning models [23, 22] as well as a learning objective [1, 2]. Fischer [7] proposes a conditional entropy bottleneck (CEB) based on IB and observes its robust generalization ability empirically. Closer to us, Alemi et al. [1] propose a variational information bottleneck (VIB) for supervised learning. They empirically show that training VIB on natural examples provides good generalization and adversarial robustness. We show that HBaR can be combined with various adversarial defense methods enhancing their robustness, but also outperforms VIB [1] when given access only to natural samples. Moreover, *we provide theoretical guarantees on how HBaR bounds the output perturbation induced by adversarial attacks.*

**Mutual Information vs. HSIC.** Mutual information is difficult to compute in practice. To address this, Alemi et al. [1] estimate IB via variational inference. Ma et al. [16] replaced mutual information by the Hilbert Schmidt Independence Criterion (HSIC) and named this the *HSIC Bottleneck* (HB). Like Ma et al. [16], we utilize HSIC to estimate IB. However, our method is different from Ma et al. [16] in several aspects. First, they use HB to train the neural network stage-wise, layer-by-layer, without backpropagation, while we use HSIC bottleneck as a regularization in addition to cross-entropy and optimize the parameters jointly by backpropagation. Second, they only evaluate the model performance on classification accuracy, while we demonstrate adversarial robustness. Finally, we show that HBaR further enhances robustness to adversarial examples both theoretically and experimentally. Greenfeld et al. [9] use HSIC between the residual of the prediction and the input data as a learning objective for model robustness on covariate distribution shifts. Their focus is on robustness to distribution shifts, whereas our work focuses on robustness to adversarial examples, on which HBaR outperforms their proposed objective.

## 3 Background

### 3.1 Adversarial Robustness

In standard $k$-ary classification, we are given a dataset $\mathcal{D} = \{(x_i, y_i)\}_{i=1}^n$, where $x_i \in \mathbb{R}^{d_X}, y_i \in \{0,1\}^k$ are i.i.d. samples drawn from joint distribution $P_{XY}$. A learner trains a neural network $h_\theta : \mathbb{R}^{d_X} \to \mathbb{R}^k$ parameterized by weights $\theta \in \mathbb{R}^{d_\theta}$ to predict $Y$ from $X$ by minimizing

$$\mathcal{L}(\theta) = \mathbb{E}_{XY}[\ell(h_\theta(X), Y)] \approx \frac{1}{n} \sum_{i=1}^n \ell(h_\theta(x_i), y_i), \tag{1}$$

where $\ell : \mathbb{R}^k \times \mathbb{R}^k \to \mathbb{R}$ is a loss function, e.g., cross-entropy. We aim to find a model $h_\theta$ that has high prediction accuracy but is also *adversarially robust*: the model should maintain high prediction accuracy against a constrained adversary, that can perturb input samples in a restricted fashion. Formally, prior to submitting a sample $x \in \mathbb{R}^{d_X}$ to the classifier, an adversary may perturb $x$ by an arbitrary $\delta \in \mathcal{S}_r$, where $\mathcal{S}_r \subseteq \mathbb{R}^{d_X}$ is the $\ell_\infty$-ball of radius $r$, i.e.,

$$\mathcal{S}_r = B(0, r) = \{\delta \in \mathbb{R}^{d_X} : \|\delta\|_\infty \leq r\}. \tag{2}$$

The *adversarial robustness* [17] of a model $h_\theta$ is measured by the expected loss attained by such adversarial examples, i.e.,

$$\mathcal{L}_r(\theta) = \mathbb{E}_{XY}\left[\max_{\delta \in \mathcal{S}_r} \ell\left(h_\theta(X + \delta), Y\right)\right] \approx \frac{1}{n} \sum_{i=1}^n \max_{\delta \in \mathcal{S}_r} \ell(h_\theta(x_i + \delta), y_i). \tag{3}$$

An adversarially robust neural network $h_\theta$ can be obtained via *adversarial training*, i.e., by minimizing the adversarial robustness loss in (3) empirically over the training set $\mathcal{D}$. In practice, this amounts to training via stochastic gradient descent (SGD) over adversarial examples $x_i + \delta$ (see, e.g., [17]). In each epoch, $\delta$ is generated on a per sample basis via an inner optimization over $\mathcal{S}_r$, e.g., via projected gradient descent (PGD) on $-\mathcal{L}$.

## 3.2 Hilbert-Schmidt Independence Criterion (HSIC)

The Hilbert-Schmidt Independence Criterion (HSIC) is a statistical dependency measure introduced by Gretton et al. [10]. HSIC is the Hilbert-Schmidt norm of the cross-covariance operator between the distributions in Reproducing Kernel Hilbert Space (RKHS). Similar to Mutual Information (MI), HSIC captures non-linear dependencies between random variables. $\text{HSIC}(X, Y)$ is defined as:

$$
\begin{aligned}
\text{HSIC}(X,Y) = {} & \mathbb{E}_{XYX'Y'}\left[k_X\left(X, X'\right)k_{Y'}\left(Y, Y'\right)\right] \\
& + \mathbb{E}_{XX'}\left[k_X\left(X, X'\right)\right]\mathbb{E}_{YY'}\left[k_Y\left(Y, Y'\right)\right] \\
& - 2\mathbb{E}_{XY}\left[\mathbb{E}_{X'}\left[k_X\left(X, X'\right)\right]\mathbb{E}_{Y'}\left[k_Y\left(Y, Y'\right)\right]\right],
\end{aligned}
\tag{4}
$$

where $X'$, $Y'$ are independent copies of $X$, $Y$, respectively, and $k_X$, $k_Y$ are kernels.

In practice, we often approximate HSIC empirically. Given $n$ i.i.d. samples $\{(x_i, y_i)\}_{i=1}^n$ drawn from $P_{XY}$, we estimate HSIC via:

$$
\widehat{\text{HSIC}}(X,Y) = (n-1)^{-2}\,\text{tr}\left(K_X H K_Y H\right),
\tag{5}
$$

where $K_X$ and $K_Y$ are kernel matrices with entries $K_{X_{ij}} = k_X(x_i, x_j)$ and $K_{Y_{ij}} = k_Y(y_i, y_j)$, respectively, and $H = \mathbf{I} - \frac{1}{n}\mathbf{1}\mathbf{1}^\top$ is a centering matrix.

## 4 Methodology

In this section, we present our method, HSIC bottleneck as regularizer (HBaR) as a means to enhance a classifier's robustness. The effect of HBaR for adversarial robustness is illustrated in Figure 1; the HSIC bottleneck penalty reduces the sensitivity of the classifier to adversarial examples. We provide a theoretical justification for this below, in Theorems 1 and 2, but also validate the efficacy of the HSIC bottleneck extensively with experiments in Section 5.

### 4.1 HSIC Bottleneck as Regularizer for Robustness

Given a feedforward neural network $h_\theta : \mathbb{R}^{d_X} \to \mathbb{R}^k$ parameterized by $\theta$ with $M$ layers, and an input r.v. $X$, we denote by $Z_j \in \mathbb{R}^{d_{Z_j}}$, $j \in \{1, \ldots, M\}$, the output of the $j$-th layer under input $X$ (i.e., the $j$-th latent representation). We define our HBaR learning objective as follows:

$$
\tilde{\mathcal{L}}(\theta) = \mathcal{L}(\theta) + \lambda_x \sum_{j=1}^{M} \text{HSIC}(X, Z_j) - \lambda_y \sum_{j=1}^{M} \text{HSIC}(Y, Z_j),
\tag{6}
$$

where $\mathcal{L}$ is the standard loss given by Eq. (1) and $\lambda_x, \lambda_y \in \mathbb{R}_+$ are balancing hyperparameters.

Together, the second and third terms in Eq. (6) form the HSIC bottleneck penalty. As HSIC measures dependence between two random variables, minimizing $\text{HSIC}(X, Z_i)$ corresponds to removing redundant or noisy information contained in $X$. Hence, this term also naturally reduces the influence of an adversarial attack, i.e., a perturbation added on the input data. This is intuitive, but we also provide theoretical justification in the next subsection. Meanwhile, maximizing $\text{HSIC}(Y, Z_i)$ encourages this lack of sensitivity to the input to happen while retaining the discriminative nature of the classifier, captured by dependence to useful information w.r.t. the output label $Y$. Note that minimizing $\text{HSIC}(X, Z_i)$ alone would also lead to the loss of useful information, so it is necessary to keep the $\text{HSIC}(Y, Z_i)$ term to make sure $Z_i$ is informative enough of $Y$.

The overall algorithm is described in Alg. 1. In practice, we perform Stochastic Gradient Descent (SGD) over $\tilde{\mathcal{L}}$: both $\mathcal{L}$ and HSIC can be evaluated empirically over batches. For the latter, we use the estimator (5), restricted over the current batch. As we have $m$ samples in a mini-batch, the complexity of calculating the empirical HSIC (5) is $O(m^2 d_{\bar{Z}})$ [24] for a single layer, where $d_{\bar{Z}} = \max_j d_{Z_j}$. Thus, the overall complexity for (6) is $O(Mm^2 d_{\bar{Z}})$. This computation is highly parallelizable, thus, the additional computation time of HBaR is small when compared to training a neural network via cross-entropy only.

**Algorithm 1:** Robust Learning with HBaR

---

**Input:** input sample tuples $\{(x_i, y_i)\}_{i=1}^n$, kernel function $k_x, k_y, k_z$, a neural network $h_\theta$
 parameterized by $\theta$, mini-batch size $m$, learning rate $\alpha$.
**Output:** parameter of classifier $\theta$
**while** *$\theta$ has not converged* **do**
 Sample a mini-batch of size $m$ from input samples.
 Forward Propagation: calculate $z_i$ and $h_\theta(x)$.
 Compute kernel matrices for $X, Y$ and $Z_i$ using $k_x, k_y, k_z$ respectively inside mini-batch.
 Compute $\tilde{\mathcal{L}}(\theta)$ via (6), where HSIC is evaluated empirically via (5).
 Backward Propagation: $\theta \leftarrow \theta - \alpha \nabla \tilde{\mathcal{L}}(\theta)$.
**end**

---

## 4.2 Combining HBaR with Adversarial Examples

HBaR can also be naturally applied in combination with adversarial training. For $r > 0$ the magnitude of the perturbations introduced in adversarial examples, one can optimize the following objective instead of $\tilde{\mathcal{L}}(\theta)$ in Eq. (6):

$$\tilde{\mathcal{L}}_r(\theta) = \mathcal{L}_r(\theta) + \lambda_x \sum_{j=1}^M \mathrm{HSIC}(X, Z_j) - \lambda_y \sum_{j=1}^M \mathrm{HSIC}(Y, Z_j), \tag{7}$$

where $\mathcal{L}_r$ is the adversarial loss given by Eq. (3). This can be used instead of $\mathcal{L}$ in Alg. 1. Adversarial examples need to be used in the computation of the gradient of the loss $\mathcal{L}_r$ in each minibatch; these need to be computed on a per sample basis, e.g., via PGD over $\mathcal{S}_r$, at additional computational cost. Note that the natural samples $(x_i, y_i)$ in a batch are used to compute the HSIC bottleneck regularizer.

The HBaR penalty can similarly be combined with other adversarial learning methods and/or used with different means for selecting adversarial examples, other than PGD. We illustrate this in Section 5, where we combine HBaR with state-of-the-art adversarial learning methods TRADES [36] and MART [30].

## 4.3 HBaR Robustness Guarantees

We provide here a formal justification for the use of HBaR to enhance robustness: we prove that regularization terms $\mathrm{HSIC}(X, Z_j), j = 1, \ldots, M$ lead to classifiers which are less sensitive to input perturbations. For simplicity, we focus on the case where $k = 1$ (i.e., binary classification). Let $Z \in \mathbb{R}^{d_Z}$ be the latent representation at some arbitrary intermediate layer of the network. That is, $Z = Z_j$, for some $j \in \{1, \ldots, M\}$; we omit the subscript $j$ to further reduce notation clutter. Then $h_\theta = (g \circ f)$, where $f : \mathbb{R}^{d_X} \to \mathbb{R}^{d_Z}$ maps the inputs to this intermediate layer, and $g : \mathbb{R}^{d_Z} \to \mathbb{R}$ maps the intermediate layer to the final layer. Then, $Z = f(X)$ and $g(Z) = h_\theta(X) \in \mathbb{R}$ are the latent and final outputs, respectively. Recall that, in HBaR, $\mathrm{HSIC}(X, Z)$ is associated with kernels $k_X, k_Z$. We make the following technical assumptions:

**Assumption 1.** *Let $\mathcal{X} \subseteq \mathbb{R}^{d_X}$, $\mathcal{Z} \subseteq \mathbb{R}^{d_Z}$ be the supports of random variables $X$, $Z$, respectively. We assume that both $h_\theta$ and $g$ are continuous and bounded functions in $\mathcal{X}$, $\mathcal{Z}$, respectively, i.e.:*

$$h_\theta \in C(\mathcal{X}), g \in C(\mathcal{Z}). \tag{8}$$

*Moreover, we assume that all functions $h_\theta$ and $g$ we consider are uniformly bounded, i.e., there exist $0 < M_\mathcal{X}, M_\mathcal{Z} < \infty$ such that:*

$$M_\mathcal{X} = \max_{h_\theta \in C(\mathcal{X})} \|h_\theta\|_\infty \quad and \quad M_\mathcal{Z} = \max_{g \in C(\mathcal{Z})} \|g\|_\infty. \tag{9}$$

The continuity stated in Assumption 1 is natural, if all activation functions are continuous. Boundedness follows if, e.g., $\mathcal{X}$, $\mathcal{Z}$ are closed and bounded (i.e., compact), or if activation functions are bounded (e.g., softmax, sigmoid, etc.).

**Assumption 2.** *We assume kernels $k_X$, $k_Z$ are universal with respect to functions $h_\theta$ and $g$ that satisfy Assumption 1, i.e., if $\mathcal{F}$ and $\mathcal{G}$ are the induced RKHSs for kernels $k_X$ and $k_Z$, respectively, then for any $h_\theta$, $g$ that satisfy Assumption 1 and any $\varepsilon > 0$ there exist functions $h' \in \mathcal{F}$ and $g' \in \mathcal{G}$ such that $\|h_\theta - h'\|_\infty \leq \varepsilon$ and $\|g - g'\|_\infty \leq \varepsilon$. Moreover, functions in $\mathcal{F}$ and $\mathcal{G}$ are uniformly bounded, i.e., there exist $0 < M_\mathcal{F}, M_\mathcal{G} < \infty$ such that for all $h' \in \mathcal{F}$ and all $g' \in \mathcal{G}$:*

$$M_\mathcal{F} = \max_{f' \in \mathcal{F}} \|f'\|_\infty \quad and \quad M_\mathcal{G} = \max_{g' \in \mathcal{G}} \|g'\|_\infty. \tag{10}$$

We note that several kernels used in practice are universal, including, e.g., the Gaussian and Laplace kernels. Moreover, given that functions that satisfy Assumption 1 are uniformly bounded by (9), such kernels can indeed remain universal while satisfying (10) via an appropriate rescaling.

Our first result shows that $\mathrm{HSIC}(X, Z)$ *at any intermediate layer $Z$* bounds the *output* variance:

**Theorem 1.** *Under Assumptions 1 and 2, we have:*

$$\mathrm{HSIC}(X, Z) \geq \frac{M_\mathcal{F} M_\mathcal{G}}{M_\mathcal{X} M_\mathcal{Z}} \sup_\theta \mathrm{Var}(h_\theta(X)). \tag{11}$$

The proof of Theorem 1 is in Appendix B in the supplement. We use a result by Greenfeld and Shalit [9] that links $\mathrm{HSIC}(X, Z)$ to the supremum of the covariance of bounded continuous functionals over $\mathcal{X}$ and $\mathcal{Z}$. Theorem 1 indicates that the regularizer $\mathrm{HSIC}(X, Z)$ at any intermediate layer naturally suppresses the variability of the output, i.e., the classifier prediction $h_\theta(X)$. To see this, observe that by Chebyshev's inequality [20] the distribution of $h_\theta(X)$ concentrates around its mean when $\mathrm{Var}(h_\theta(X))$ approaches $0$. As a result, bounding $\mathrm{HSIC}(X, Z)$ inherently also bounds the (global) variability of the classifier (across all parameters $\theta$). This observation motivates us to also maximize $\mathrm{HSIC}(Y, Z)$ to recover essential information useful for classification: if we want to achieve good adversarial robustness as well as good predictive accuracy, we have to strike a balance between $\mathrm{HSIC}(X, Z)$ and $\mathrm{HSIC}(Y, Z)$. This perfectly aligns with the intuition behind the information bottleneck [28] and the well-known accuracy-robustness trade off [17, 36, 29, 21]. We also confirm this experimentally: we observe that both additional terms (the standard loss and $\mathrm{HSIC}(Y, Z)$) are necessary for ensuring good prediction performance in practice (see Table 3).

Most importantly, by further assuming that features are normal, we can show that HSIC bounds the power of an arbitrary adversary, as defined in Eq. (3):

**Theorem 2.** *Assume that $X \sim \mathcal{N}(0, \sigma^2 \mathbf{I})$. Then, under Assumptions 1 and 2, we have:*[3]

$$\frac{r\sqrt{-2\log o(1)}d_X M_\mathcal{Z}}{\sigma M_\mathcal{F} M_\mathcal{G}} \mathrm{HSIC}(X, Z) + o(r) \geq \mathbb{E}[|h_\theta(X + \delta) - h_\theta(X)|], \quad for\ all\ \delta \in \mathcal{S}_r. \tag{12}$$

The proof of Theorem 2 can also be found in Appendix C in the supplement. We again use the result by Greenfeld and Shalit [9] along with Stein's Lemma [15], that relates covariances of Gaussian r.v.s and their functions to expected gradients. In particular, we apply Stein's Lemma to the bounded functionals considered by Greenfeld and Shalit by using a truncation argument. Theorem 2 implies that $\mathrm{HSIC}(X, Z)$ indeed bounds the output perturbation produced by an arbitrary adversary: suppressing HSIC sufficiently can ensure that the adversary cannot alter the output significantly, in expectation. In particular, if $\mathrm{HSIC}(X, Z) = o\left(\frac{\sigma M_\mathcal{F} M_\mathcal{G}}{\sqrt{-2\log o(1)}d_X M_\mathcal{Z}}\right)$, then $\lim_{r \to 0} \sup_{\delta \in \mathcal{S}_r} \mathbb{E}[|h_\theta(X + \delta) - h_\theta(X)|]/r = 0$, i.e., the output is almost constant under small input perturbations.

## 5 Experiments

### 5.1 Experimental Setting

We experiment with three standard datasets, MNIST [14], CIFAR-10 [12] and CIFAR-100 [12]. We use a 4-layer LeNet [17] for MNIST, ResNet-18 [11] and WideResNet-28-10 [35] for CIFAR-10,

---

[3]Recall that for functions $f, g : \mathbb{R} \to \mathbb{R}$ we have $f = o(g)$ if $\lim_{r \to 0} \frac{f(r)}{g(r)} = 0$.

Table 1: Natural test accuracy (in %), adversarial robustness ((in %) on FGSM, PGD, CW, and AA attacked test examples) on MNIST and CIFAR-100 of **[row i, iii, v] adversarial learning baselines** and **[row ii, iv, vi] combining HBaR with each correspondingly**. Each result is the average of five runs.

| Methods | MNIST by LeNet | | | | | | CIFAR-100 by WideResNet-28-10 | | | | | |
| --- | --- | --- | --- | --- | --- | --- | --- | --- | --- | --- | --- | --- |
| | Natural | FGSM | PGD$^{20}$ | PGD$^{40}$ | CW | AA | Natural | FGSM | PGD$^{10}$ | PGD$^{20}$ | CW | AA |
| PGD | 98.40 | 93.44 | 94.56 | 89.63 | 91.20 | 86.62 | 59.91 | 29.85 | 26.05 | 25.38 | 22.28 | 20.91 |
| HBaR + PGD | **98.66** | **96.02** | **96.44** | **94.35** | **95.10** | **91.57** | **63.84** | **31.59** | **27.90** | **27.21** | **23.23** | **21.61** |
| TRADES | **97.64** | 94.73 | 95.05 | 93.27 | 93.05 | 89.66 | 60.29 | 34.19 | 31.32 | 30.96 | 28.20 | 26.91 |
| HBaR + TRADES | **97.64** | **95.23** | **95.17** | **93.49** | **93.47** | **89.99** | **60.55** | **34.57** | **31.96** | **31.57** | **28.72** | **27.46** |
| MART | **98.29** | 95.57 | 95.23 | 93.55 | 93.45 | 88.36 | 58.42 | 32.94 | 29.17 | 28.19 | 27.31 | 25.09 |
| HBaR + MART | 98.23 | **96.09** | **96.08** | **94.64** | **94.62** | **89.99** | **58.93** | **33.49** | **30.72** | **30.16** | **28.89** | **25.21** |

Table 2: Natural test accuracy (in %), adversarial robustness ((in %) on FGSM, PGD, CW, and AA attacked test examples) on CIFAR-10 by ResNet-18 and WideResNet-28-10 of **[row i, iii, v] adversarial learning baselines** and **[row ii, iv, vi] combining HBaR with each correspondingly**. Each result is the average of five runs.

| Methods | CIFAR-10 by ResNet-18 | | | | | | CIFAR-10 by WideResNet-28-10 | | | | | |
| --- | --- | --- | --- | --- | --- | --- | --- | --- | --- | --- | --- | --- |
| | Natural | FGSM | PGD$^{10}$ | PGD$^{20}$ | CW | AA | Natural | FGSM | PGD$^{10}$ | PGD$^{20}$ | CW | AA |
| PGD | 84.71 | 55.95 | 49.37 | 47.54 | 41.17 | 43.42 | 86.63 | 58.53 | 52.21 | 50.59 | 49.32 | 47.25 |
| HBaR + PGD | **85.73** | **57.13** | **49.63** | **48.32** | **41.80** | **44.46** | **87.91** | **59.69** | **52.72** | **51.17** | **49.52** | **47.60** |
| TRADES | 84.07 | 58.63 | 53.21 | 52.36 | 50.07 | 49.38 | **85.66** | 61.55 | 56.62 | 55.67 | 54.02 | 52.71 |
| HBaR + TRADES | **84.10** | **58.97** | **53.76** | **52.92** | **51.00** | **49.43** | 85.61 | **62.20** | **57.30** | **56.51** | **54.89** | **53.53** |
| MART | 82.15 | 59.85 | 54.75 | 53.67 | 50.12 | 47.97 | **85.94** | 59.39 | 51.30 | 49.46 | 47.94 | 45.48 |
| HBaR + MART | **82.44** | **59.86** | **54.84** | **53.89** | **50.53** | **48.21** | 85.52 | **60.54** | **53.42** | **51.81** | **49.32** | **46.99** |

and WideResNet-28-10 [35] for CIFAR-100. We use cross-entropy as loss $\mathcal{L}(\theta)$. Licensing information for all existing assets can be found in Appendix D in the supplement.

**Algorithms.** We compare *HBaR* to the following non-adversarial learning algorithms: *Cross-Entropy (CE)*, *Stage-Wise HSIC Bottleneck (SWHB)* [16], *XIC* [9], and *Variational Information Bottleneck (VIB)* [1]. We also incorporate HBaR to several adversarial learning algorithms, as described in Section 4.2, and compare against the original methods, without the HBaR penalty. The adversarial methods we use are: *Projected Gradient Descent (PGD)* [17], *TRADES* [36], and *MART* [30]. Further details and parameters can be found in Appendix E in the supplement.

**Performance Metrics.** For all methods, we evaluate the obtained model $h_\theta$ via the following metrics: (a) *Natural* (i.e., clean test data) accuracy, and adversarial robustness via test accuracy under (b) *FGSM*, the fast gradient sign attack [8], (c) *PGD$^m$*, the PGD attack with $m$ steps used for the internal PGD optimization [17], (d) *CW*, the CW-loss within the PGD framework [3], and (e) *AA*, AutoAttack [5]. All five metrics are reported in percent (%) accuracy. Following prior literature, we set step size to 0.01 and radius $r = 0.3$ for MNIST, and step size as $2/255$ and $r = 8/255$ for CIFAR-10 and CIFAR-100. All attacks happen during the test phase and have full access to model parameters (i.e., are white-box attacks). All experiments are carried out on a Tesla V100 GPU with 32 GB memory and 5120 cores.

## 5.2 Results

**Combining HBaR with Adversarial Examples.** We show how HBaR can be used to improve robustness when used as a regularizer, as described in Section 4.2, along with state-of-the-art adversarial learning methods. We run each experiment by five times and report the mean natural test accuracy and adversarial robustness of all models on MNIST, CIFAR-10, and CIFAR-100 datasets by four architectures in Table 1 and Table 2. Combined with all adversarial training baselines, HBaR *consistently improves adversarial robustness against all types of attacks on all datasets*. The resulting improvements are larger than 2 standard deviations (that range between 0.05-0.2) in most cases; we report the results with standard deviations in Appendix G in the supplement. Although natural accuracy is generally restricted by the trade-off between robustness and accuracy [36], we observe that incorporating HBaR comes with an actual improvement over natural accuracy in most cases.

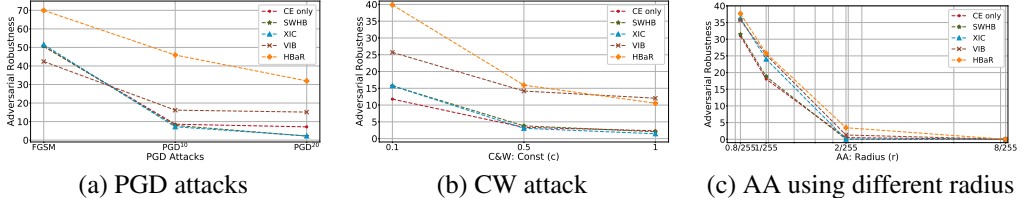

| (a) PGD attacks | (b) CW attack | (c) AA using different radius |

Figure 2: CIFAR-10 by ResNet-18: Adversarial robustness of **IB-based baselines** and **proposed HBaR** under (a) PGD attacks, (b) CW attack by various of constant $c$, and (c) AA using different radius. Interestingly, while achieving the highest adversarial robustness under almost all cases, HBaR achieves natural accuracy (95.27%) comparable to CE (95.32%) which is much higher than VIB (92.35%), XIC (92.93%) and SWHB (59.18%).

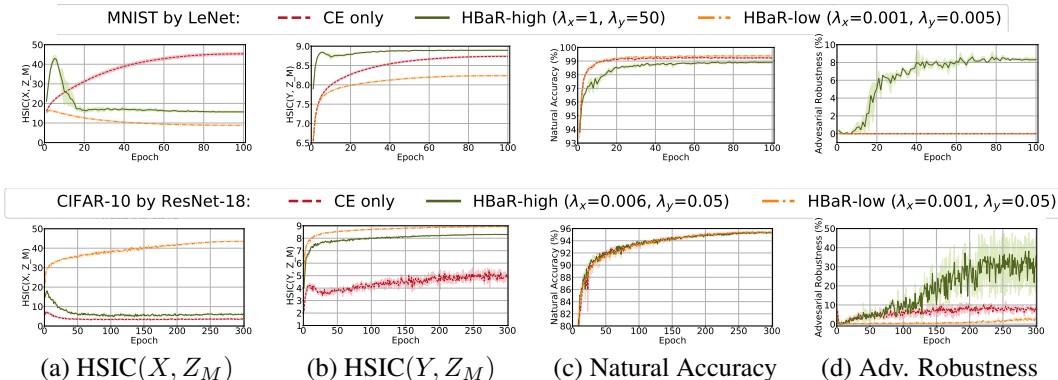

| (a) HSIC$(X, Z_M)$ | (b) HSIC$(Y, Z_M)$ | (c) Natural Accuracy | (d) Adv. Robustness |

Figure 3: Visualization of the HBaR quantities (a) HSIC$(X, Z_M)$, (b) HSIC$(X, Z_M)$, (c) natural test accuracy, and (d) adversarial robustness against PGD attack (PGD$^{40}$ and PGD$^{20}$ on MNIST and CIFAR-10, respectively) as a function of training epochs, on MNIST by LeNet (top) and CIFAR-10 by ResNet (bottom). Different colored lines correspond to CE, HBaR-high (HBaR with high weights $\lambda$), and HBaR-low (HBaR method small weighs $\lambda$). HBaR-low parameters are selected so that the values of the loss $\mathcal{L}$ and each of the HSIC terms are close after the first epoch.

**Adversarial Robustness Analysis without Adversarial Training.** Next, we show that HBaR can achieve modest robustness even without adversarial examples during training. We evaluate the robustness of HBaR on CIFAR-10 by ResNet-18 against various adversarial attacks, and compare HBaR with other information bottleneck penalties without adversarial training in Figure 2. Specifically, we compare the robustness of HBaR with other IB-based methods under various attacks and hyperparameters. Our proposed HBaR achieves the best overall robustness against all three types of attacks while attaining competitive natural test accuracy. Interestingly, HBaR achieves natural accuracy (95.27%) comparable to CE (95.32%) which is much higher than VIB (92.35%), XIC (92.93%) and SWHB (59.18%). We observe SWHB underperforms HBaR on CIFAR-10 for both natural accuracy and robustness. One possible explanation may be that when the model is deep, minimizing HSIC without backpropagation, as in SWHB, does not suffice to transmit the learned information across layers. Compared to SWHB, HBaR backpropagates over the HSIC objective through each intermediate layer and computes gradients only once in each batch, improving accuracy and robustness while reducing computational cost significantly.

**Synergy between HSIC Terms.** Focusing on $Z_M$, the last latent layer, Figure 3 shows the evolution per epoch of: (a) HSIC$(X, Z_M)$, (b) HSIC$(Y, Z_M)$, (c) natural accuracy (in %), and (d) adversarial robustness (in %) under PGD attack on MNIST and CIFAR-10. Different lines correspond to CE, HBaR-high (HBaR with high weights $\lambda$), and HBaR-low (HBaR method small weighs $\lambda$). HBaR-low parameters are selected so that the values of the loss $\mathcal{L}$ and each of the HSIC terms are close after the first epoch. Figure 3(c) illustrates that all three settings achieve good natural accuracy on both datasets. However, in Figure 3(d), only HBaR-high, that puts sufficient weight on HSIC terms, attains relatively high adversarial robustness. In Figure 3(a), we see that CE leads to high HSIC$(X, Z_M)$ for the shallow LeNet, but low in the (much deeper) ResNet-18, even lower than HBaR-low. Moreover, we also see that the best performer in terms of adversarial robustness, HBaR-high, lies in between the other two w.r.t. HSIC$(X, Z_M)$. Both of these observations indi-

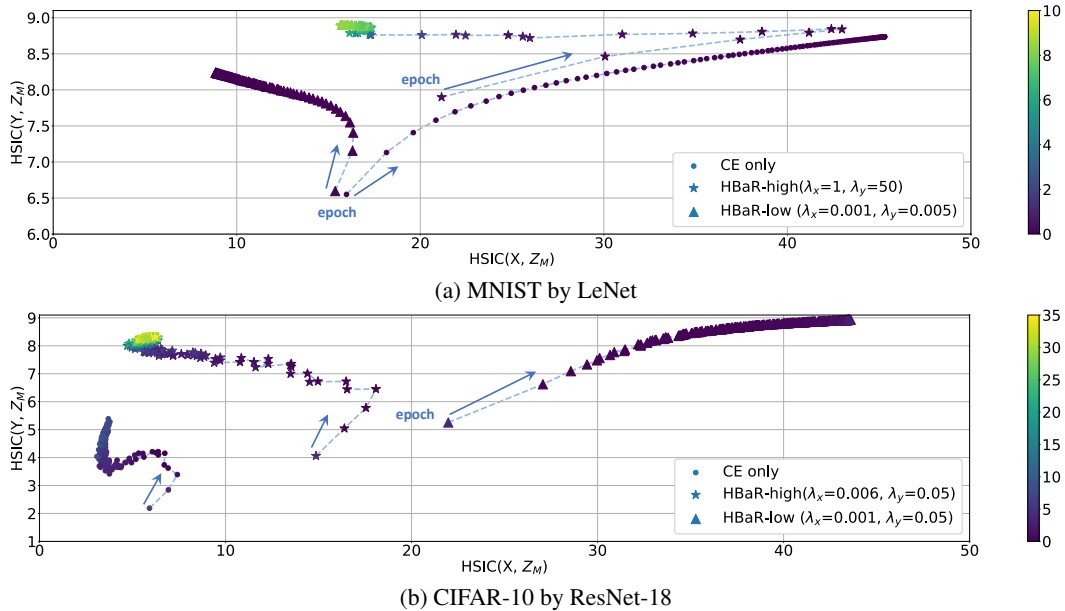

(a) MNIST by LeNet

(b) CIFAR-10 by ResNet-18

Figure 4: HSIC plane dynamics versus adversarial robustness. The x-axis plots HSIC between the last intermediate layer $Z_M$ and the input $X$, while the y-axis plots HSIC between $Z_M$ and the output $Y$. The color scale indicates adversarial robustness against PGD attack ($PGD^{40}$ and $PGD^{20}$ on MNIST and CIFAR-10, respectively). The arrows indicate dynamic direction w.r.t. training epochs. Each marker in the figures represents a different setting: **dots**, **stars**, and **triangles** represent CE-only, HBaR-high, and HBaR-low, respectively, compatible with the definition in Figure 3.

cate the importance of the $\mathrm{HSIC}(Y, Z_M)$ penalty: minimizing $\mathrm{HSIC}(X, Z_M)$ appropriately leads to good adversarial robustness, but coupling learning to labels via the third term is integral to maintaining useful label-related information in latent layers, thus resulting in good adversarial robustness. Figure 3(b) confirms this, as HBaR-high achieves relatively high $\mathrm{HSIC}(Y, Z_M)$ on both datasets.

Figure 4 provides another perspective of the same experiments via the learning dynamics on the HSIC plane. We again observe that the best performer in terms of robustness HBaR-high lies in between the other two methods, crucially attaining a much higher $\mathrm{HSIC}(Y, Z_M)$ than HBaR-low. Moreover, for both HBaR methods, we clearly observe the two distinct optimization phases first observed by Shwartz-Ziv and Tishby [23] in the context of the mutual information bottleneck: the *fast empirical risk minimization phase*, where the neural network tries to learn a meaningful representation by increasing $\mathrm{HSIC}(Y, Z_M)$ regardless of information redundancy ($\mathrm{HSIC}(X, Z_M)$ increasing), and the *representation compression phase*, where the neural network turns its focus onto compressing the latent representation by minimizing $\mathrm{HSIC}(X, Z_M)$, while maintaining highly label-related information. Interestingly, the HBaR penalty produces the two-phase behavior even though our networks use ReLU activation functions; Shwartz et al. [23] only observed these two optimization phases on neural networks with tanh activation functions, a phenomenon further confirmed by Saxe et al. [22].

**Ablation Study.** Motivated by the above observations, we turn our attention to how the three terms in the loss function in Eq. (6) affect HBaR. As illustrated in Table 3, removing any part leads to either a significant natural accuracy or robustness degradation. Specifically, using $\mathcal{L}(\theta)$ only (row [i]) lacks adversarial robustness; removing $\mathcal{L}(\theta)$ (row [ii]) or the penalty on $Y$ (row [iii]) degrades natural accuracy significantly (a similar result was also observed in [2]); finally, removing the penalty on $X$ improves the natural accuracy while degrading adversarial robustness. The three terms combined together by proper hyperparameters $\lambda_x$ and $\lambda_y$ (row [v]) achieve both high natural accuracy and adversarial robustness. We provide a comprehensive ablation study on the sensitivity of $\lambda_x$ and $\lambda_y$ and draw conclusions in Appendix F in the supplement (Tables 7 and 8).

Table 3: Ablation study on HBaR. Rows [i-iv] indicate the effect of removing each component of the learning objective defined in Eq.(6) (row [v]). We evaluate each objective over $\text{HSIC}(X, Z_M)$, $\text{HSIC}(Y, Z_M)$, natural test accuracy (in %), and adversarial robustness (in %) against PGD[40] and PGD[20] on MNIST and CIFAR-10 respectively. We set $\lambda_x$ as 1 and 0.006, $\lambda_y$ as 50 and 0.05, for MNIST and CIFAR-10 respectively.

| Rows | Objectives | MNIST by LeNet | | | | CIFAR-10 by ResNet-18 | | | |
| --- | --- | --- | --- | --- | --- | --- | --- | --- | --- |
| | | HSIC | | Natural | PGD[40] | HSIC | | Natural | PGD[20] |
| | | $(X, Z_M)$ | $(Y, Z_M)$ | | | $(X, Z_M)$ | $(Y, Z_M)$ | | |
| [i] | $\mathcal{L}(\theta)$ | 45.29 | 8.73 | 99.23 | 0.00 | 3.45 | 4.76 | 95.32 | 8.57 |
| [ii] | $\lambda_x \sum_j \text{HSIC}(X, Z_j) - \lambda_y \sum_j \text{HSIC}(Y, Z_j)$ | 16.45 | 8.65 | 30.08 | 9.47 | 44.37 | 8.72 | 19.30 | 8.58 |
| [iii] | $\mathcal{L}(\theta) + \lambda_x \sum_j \text{HSIC}(X, Z_j)$ | 0.00 | 0.00 | 11.38 | 10.00 | 0.00 | 0.00 | 10.03 | 10.10 |
| [iv] | $\mathcal{L}(\theta) - \lambda_y \sum_j \text{HSIC}(Y, Z_j)$ | 56.38 | 9.00 | 99.33 | 0.00 | 43.71 | 8.93 | 95.50 | 1.90 |
| [v] | $\mathcal{L}(\theta) + \lambda_x \sum_j \text{HSIC}(X, Z_j) - \lambda_y \sum_j \text{HSIC}(Y, Z_j)$ | 15.68 | 8.89 | 98.90 | 8.33 | 6.07 | 8.30 | 95.35 | 34.85 |

# 6 Conclusions

We investigate the HSIC bottleneck as regularizer (HBaR) as a means to enhance adversarial robustness. We theoretically prove that HBaR suppresses the sensitivity of the classifier to adversarial examples while retaining its discriminative nature. One limitation of our method is that the robustness gain is modest when training with only natural examples. Moreover, a possible negative societal impact is overconfidence in adversarial robustness: over-confidence in the *adversarially-robust* models produced by HBaR as well as other defense methods may lead to overlooking their potential failure on newly-invented attack methods; this should be taken into account in safety-critical applications like healthcare [6] or security [26]. We extend the discussion on the limitations and potential negative societal impacts of our work in Appendix H and I, respectively, in the supplement.

# 7 Acknowledgements

The authors gratefully acknowledge support by the National Science Foundation under grants CCF-1937500 and CNS-2112471, and the National Institutes of Health under grant NHLBI U01HL089856.

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
