# B  Proof of Theorem 1

*Proof.* The following lemma holds:

**Lemma 3.** *[10, 9] Let $X$, $Z$ be random variables residing in metric spaces $\mathcal{X}$, $\mathcal{Z}$, respectively. Let also $\mathcal{F}, \mathcal{G}$ be the two separable RKHSs on $\mathcal{X}, \mathcal{Z}$ induced by $k_X$ and $k_Z$, respectively. Then, the following inequality holds:*

$$\mathrm{HSIC}(X, Z) \geq \sup_{s \in \mathcal{F}, t \in \mathcal{G}} \mathrm{Cov}[s(X), t(Z)]. \tag{13}$$

Lemma 3 shows that HSIC bounds the supremum of the covariance between any pair of functions in the RKHS, $\mathcal{F}, \mathcal{G}$. Assumption 2 states that functions in $\mathcal{F}$ and $\mathcal{G}$ are uniformly bounded by $M_\mathcal{F} > 0$ and $M_\mathcal{G} > 0$, respectively. Let $\tilde{\mathcal{F}}$ and $\tilde{\mathcal{G}}$ be the restriction of $\mathcal{F}$ and $\mathcal{G}$ to functions in the unit ball of the respective RKHSs through rescaling, i.e.:

$$\tilde{\mathcal{F}} = \left\{ \frac{h}{M_\mathcal{F}} : h \in \mathcal{F} \right\} \quad \text{and} \quad \tilde{\mathcal{G}} = \left\{ \frac{g}{M_\mathcal{G}} : g \in \mathcal{G} \right\}. \tag{14}$$

The following lemma links the covariance of the functions in the original RKHSs to their normalized version:

**Lemma 4.** *[9] Suppose $\mathcal{F}$ and $\mathcal{G}$ are RKHSs over $\mathcal{X}$ and $\mathcal{Z}$, s.t. $\|s\|_\infty \le M_\mathcal{F}$ for all $s \in \mathcal{F}$ and $\|t\|_\infty \le M_\mathcal{G}$ for all $t \in \mathcal{G}$. Then the following holds:*

$$\sup_{s \in \mathcal{F}, t \in \mathcal{G}} \text{Cov}[s(X), t(Z)] = M_\mathcal{F} M_\mathcal{G} \sup_{s \in \tilde{\mathcal{F}}, t \in \tilde{\mathcal{G}}} \text{Cov}[s(X), t(Z)]. \tag{15}$$

For simplicity in notation, we define the following sets containing functions that satisfy Assumption 1:

$$C_b(\mathcal{X}) = \{h \in C(\mathcal{X}) : \|h\|_\infty \le M_\mathcal{X}\} \quad \text{and} \quad C_b(\mathcal{Z}) = \{g \in C(\mathcal{Z}) : \|g\|_\infty \le M_\mathcal{Z}\}. \tag{16}$$

In Assumption 2, we mention that functions in $\mathcal{F}$ and $\mathcal{G}$ may require appropriate rescaling to keep the universality of corresponding kernels. To make the rescaling explicit, we define the following *rescaled* RKHSs:

$$\hat{\mathcal{F}} = \left\{ \frac{M_\mathcal{X}}{M_\mathcal{F}} \cdot h : h \in \mathcal{F} \right\} \quad \text{and} \quad \hat{\mathcal{G}} = \left\{ \frac{M_\mathcal{Z}}{M_\mathcal{G}} \cdot g : g \in \mathcal{G} \right\}. \tag{17}$$

This rescaling ensures that $\|\hat{h}\|_\infty \le M_\mathcal{X}$ for every $\hat{h} \in \hat{\mathcal{F}}$. Similarly, $\|\hat{g}\|_\infty \le M_\mathcal{Z}$ for every $\hat{g} \in \hat{\mathcal{G}}$.

We also want to prove $\mathcal{F}$ is convex. Given $f, g \in \mathcal{F}$, we need to show for all $0 \le \alpha \le 1$, the function $\alpha f + (1-\alpha)g \in \mathcal{F}$. As linear summation of RKHS functions is in the RKHS, we just need to check that $\|\alpha f + (1-\alpha)g\|_\infty \le M_\mathcal{F}$; indeed:

$$\|\alpha f + (1-\alpha)g\|_\infty \le \alpha\|f\|_\infty + (1-\alpha)\|g\|_\infty \le \alpha M_\mathcal{F} + (1-\alpha)M_\mathcal{F} \tag{18}$$

We thus conclude that the bounded RKHS $\mathcal{F}$ is indeed convex. Hence any rescaling of the function, as long as it has a norm less than $M_\mathcal{F}$, remains inside $\mathcal{F}$.

Indeed, the following lemma holds:

**Lemma 5.** *If $\mathcal{F}, \mathcal{G}$ are universal with respect to $C_b(\mathcal{X}), C_b(\mathcal{Z})$, then:*

$$\hat{\mathcal{F}} = C_b(\mathcal{X}) \quad \text{and} \quad \hat{\mathcal{G}} = C_b(\mathcal{Z}). \tag{19}$$

*Proof.* We prove this by first showing $C_b(\mathcal{X}) \subseteq \hat{\mathcal{F}}$ and then $\hat{\mathcal{F}} \subseteq C_b(\mathcal{X})$, which leads to equality of the sets.

- $C_b(\mathcal{X}) \subseteq \hat{\mathcal{F}}$: For all $h \in C_b(\mathcal{X})$, we show $h \in \hat{\mathcal{F}}$. Based on the definition of $C_b(\mathcal{X})$ in (16), we know $\|h\|_\infty \le M_\mathcal{X}$. From universality stated in Assumption 2, $h \in \mathcal{F}$. Let $g = \frac{M_\mathcal{F}}{M_\mathcal{X}}h$. Then $\|g\|_\infty = \|\frac{M_\mathcal{F}}{M_\mathcal{X}}h\|_\infty = \frac{M_\mathcal{F}}{M_\mathcal{X}}\|h\|_\infty \le M_\mathcal{F}$. Based on the convexity of $\mathcal{F}$, $g \in \mathcal{F}$. We rescale every function in $\mathcal{F}$ by $\frac{M_\mathcal{X}}{M_\mathcal{F}}$ to form $\hat{\mathcal{F}}$, so $\frac{M_\mathcal{X}}{M_\mathcal{F}}g = \frac{M_\mathcal{X}}{M_\mathcal{F}}\frac{M_\mathcal{F}}{M_\mathcal{X}}h = h \in \hat{\mathcal{F}}$.

- $\hat{\mathcal{F}} \subseteq C_b(\mathcal{X})$: On the other hand, for all $h \in \hat{\mathcal{F}}$, $h$ is continuous and bounded by $M_\mathcal{X}$. So based on the definition of $C_b(\mathcal{X})$ in (16), $h \in C_b(\mathcal{X})$. Thus, $\hat{\mathcal{F}} \subseteq C_b(\mathcal{X})$.

Having both side of the inclusion we conclude that $\hat{\mathcal{F}} = C_b(\mathcal{X})$. One can prove $\hat{\mathcal{G}} = C_b(\mathcal{Z})$ similarly.

$\square$

Applying the universality of kernels from Assumption 2 we can prove the following lemma:

**Lemma 6.** *Let $X$, $Z$ be random variables residing in metric spaces $\mathcal{X}$, $\mathcal{Z}$ with separable RKHSs $\mathcal{F}$, $\mathcal{G}$ induced by kernel functions $k_X$ and $k_Z$, respectively, for which Assumption 2 holds. Let $\hat{\mathcal{F}}$ and $\hat{\mathcal{G}}$ be the rescaled RKHSs defined in* (17). *Then:*

$$\frac{M_\mathcal{X} M_\mathcal{Z}}{M_\mathcal{F} M_\mathcal{G}} \sup_{s \in \mathcal{F}, t \in \mathcal{G}} \mathrm{Cov}[s(X), t(Z)] = \sup_{s \in \hat{\mathcal{F}}, t \in \hat{\mathcal{G}}} \mathrm{Cov}[s(X), t(Z)] = \sup_{s \in C_b(\mathcal{X}), t \in C_b(\mathcal{Z})} \mathrm{Cov}[s(X), t(Z)],$$

(20)

*where $C_b(\mathcal{X}), C_b(\mathcal{Z})$ are defined in* (16).

*Proof.* The right equality of Lemma 6 immediately follows by Lemma 5:

$$\sup_{s \in \hat{\mathcal{F}}, t \in \hat{\mathcal{G}}} \mathrm{Cov}[s(X), t(Z)] = \sup_{s \in C_b(\mathcal{X}), t \in C_b(\mathcal{Z})} \mathrm{Cov}[s(X), t(Z)].$$

(21)

Applying Lemma 4 on $\mathcal{F}, \mathcal{G}, \tilde{\mathcal{F}}, \tilde{\mathcal{G}}$, we have:

$$\sup_{s \in \mathcal{F}, t \in \mathcal{G}} \mathrm{Cov}[s(X), t(Z)] = M_\mathcal{F} M_\mathcal{G} \sup_{s \in \tilde{\mathcal{F}}, t \in \tilde{\mathcal{G}}} \mathrm{Cov}[s(X), t(Z)].$$

(22)

Note that from (17) and (14), we have that the corresponding normalized space for $\hat{\mathcal{F}}$ is:

$$\left\{ \frac{h}{M_\mathcal{X}} : h \in \hat{\mathcal{F}} \right\} = \left\{ \frac{M_\mathcal{X}}{M_\mathcal{F}} \frac{h}{M_\mathcal{X}} : h \in \mathcal{F} \right\} = \left\{ \frac{h}{M_\mathcal{F}} : h \in \mathcal{F} \right\} = \tilde{\mathcal{F}}.$$

(23)

Similarly, the normalized space for $\hat{\mathcal{G}}$ is:

$$\left\{ \frac{g}{M_\mathcal{Z}} : g \in \hat{\mathcal{G}} \right\} = \left\{ \frac{g}{M_\mathcal{G}} : g \in \mathcal{G} \right\} = \tilde{\mathcal{G}}.$$

(24)

Equation (23) implies that the normalized space induced from $\hat{\mathcal{F}}$ coincides with the normalized space induced from $\mathcal{F}$. Similarly, Equation (24) implies the normalized spaces for $\mathcal{G}$ and $\hat{\mathcal{G}}$ also coincide. Moreover, for all $\hat{h} \in \hat{\mathcal{F}}, \|\hat{h}\|_\infty \leq M_\mathcal{X}$ and for all $\hat{g} \in \hat{\mathcal{G}}, \|\hat{g}\|_\infty \leq M_\mathcal{Z}$. Hence, applying Lemma 4 on $\hat{\mathcal{F}}, \hat{\mathcal{G}}, \tilde{\mathcal{F}}, \tilde{\mathcal{G}}$, we have:

$$\sup_{s \in \hat{\mathcal{F}}, t \in \hat{\mathcal{G}}} \mathrm{Cov}[s(X), t(Z)] = M_\mathcal{X} M_\mathcal{Z} \sup_{s \in \tilde{\mathcal{F}}, t \in \tilde{\mathcal{G}}} \mathrm{Cov}[s(X), t(Z)].$$

(25)

By dividing Equation (22) and (25), we prove the left part of Lemma 6:

$$\frac{M_\mathcal{X} M_\mathcal{Z}}{M_\mathcal{F} M_\mathcal{G}} \sup_{s \in \mathcal{F}, t \in \mathcal{G}} \mathrm{Cov}[s(X), t(Z)] = \sup_{s \in \hat{\mathcal{F}}, t \in \hat{\mathcal{G}}} \mathrm{Cov}[s(X), t(Z)].$$

(26)

$\square$

By combining Theorem 3 and Lemma 6, we have the following result:

$$\frac{M_\mathcal{X} M_\mathcal{Z}}{M_\mathcal{F} M_\mathcal{G}} \mathrm{HSIC}(X, Z) \geq \sup_{s \in C_b(\mathcal{X}), t \in C_b(\mathcal{Z})} \mathrm{Cov}[s(X), t(Z)].$$

(27)

Recall that $h_\theta$ is a neural network from $\mathcal{X}$ to $\mathcal{Y}$, such that it can be written as composition of $g \circ f$, where $f : \mathcal{X} \to \mathcal{Z}$ and $g : \mathcal{Z} \to \mathcal{Y}$. Moreover, $h_\theta \in C_b(\mathcal{X})$ and $g \in C_b(\mathcal{Z})$. Using the fact that the supremum on a subset of a set is smaller or equal than the supremum on the whole set, we conclude that:

$$\begin{aligned}
\frac{M_\mathcal{X} M_\mathcal{Z}}{M_\mathcal{F} M_\mathcal{G}} \mathrm{HSIC}(X, Z) &\geq \sup_\theta \mathrm{Cov}[h_\theta(X), g(Z))] \\
&= \sup_\theta \mathrm{Cov}[h_\theta(X), g \circ f(X)] \\
&= \sup_\theta \mathrm{Var}[h_\theta(X)].
\end{aligned}$$

(28)

$\square$

## C  Proof of Theorem 2

*Proof.* Let $t_i : \mathbb{R}^{d_X} \to \mathbb{R}$, $i = 1, 2, ..., d_X$ be the following truncation functions:

$$t_i(X) = \begin{cases} -R, & \text{if } X_i < -R, \\ X_i, & \text{if } -R \leq X_i \leq R, \\ R, & \text{if } X_i > R. \end{cases} \tag{29}$$

where $0 < R < \infty$ and $X_i$ is the $i$-th dimension of $X$. Functions $t_i$ are continous and bounded in $\mathcal{X}$, and

$$t_i \in C_{b'}(\mathcal{X}), \quad \text{where} \quad C_{b'}(\mathcal{X}) = \{t \in C(\mathcal{X}) : \|t\|_\infty \leq R\} \tag{30}$$

Moreover, $g$ satisfies Assumptions 1 and 2. Similar to the proof of Theorem 1, by combining Theorem 3 and Lemma 6, we have that:

$$\frac{RM_{\mathcal{Z}}}{M_{\mathcal{F}} M_{\mathcal{G}}} \text{HSIC}(X, Z) \geq \sup_{t \in C_{b'}(\mathcal{X}), \, g \in C_b(\mathcal{Z})} \text{Cov}[t(X), g(Z)]$$
$$\geq \text{Cov}[t_i(X), h_\theta(X)], \quad i = 1, \ldots, d_X. \tag{31}$$

Moreover, the following lemma holds:

**Lemma 7.** *Let $X \sim \mathcal{N}(0, \sigma^2 \mathbf{I})$ and $t_i(X)$ defined by (29). For all $h_\theta$ that satisfy Assumption 1, we have:*

$$\text{Cov}[X_i, h_\theta(X)] - \text{Cov}[t_i(X), h_\theta(X)] \leq \frac{2M_{\mathcal{X}} \sigma}{\sqrt{2\pi}} \exp(-\frac{R^2}{2\sigma^2}), \quad \text{for all } i = 1, 2, \ldots, d_X. \tag{32}$$

*Proof.*

$$\text{LHS} = \int_{-\infty}^{\infty} (x_i - t_i(x)) h_\theta(x) \frac{1}{\sqrt{2\pi\sigma^2}} \exp(-\frac{x_i^2}{2\sigma^2}) dx_i \tag{33a}$$

$$= \frac{1}{\sqrt{2\pi\sigma^2}} \left( \int_{-\infty}^{-R} (x_i + R) h_\theta(x) \exp(-\frac{x_i^2}{2\sigma^2}) dx_i + \int_{R}^{\infty} (x_i - R) h_\theta(x) \exp(-\frac{x_i^2}{2\sigma^2}) dx_i \right) \tag{33b}$$

$$\leq \frac{2M_{\mathcal{X}}}{\sqrt{2\pi\sigma^2}} \int_{R}^{\infty} (x_i - R) \exp(-\frac{x_i^2}{2\sigma^2}) dx_i \tag{33c}$$

$$= \frac{2M_{\mathcal{X}}}{\sqrt{2\pi\sigma^2}} \int_{R}^{\infty} x_i \exp(-\frac{x_i^2}{2\sigma^2}) dx_i - \frac{2M_{\mathcal{X}} R}{\sqrt{2\pi\sigma^2}} \int_{R}^{\infty} \exp(-\frac{x_i^2}{2\sigma^2}) dx_i \tag{33d}$$

$$\leq \frac{2M_{\mathcal{X}}}{\sqrt{2\pi\sigma^2}} \int_{R}^{\infty} x_i \exp(-\frac{x_i^2}{2\sigma^2}) dx_i \tag{33e}$$

$$= \frac{2M_{\mathcal{X}} \sigma}{\sqrt{2\pi}} \exp(-\frac{R^2}{2\sigma^2}), \tag{33f}$$

where (33a), (33b), (33d), (33f) are direct results from definition or simple calculation, (33c) comes from the fact that $M_{\mathcal{X}} = \max \|h_\theta(X)\|_\infty$ and the symmetry of two integrals, and (33e) is due to the non-negativity of the probability density function. $\square$

Combining Lemma 7 with (31), we have the following result:

$$\frac{RM_{\mathcal{Z}}}{M_{\mathcal{F}} M_{\mathcal{G}}} \text{HSIC}(X, Z) + \frac{2M_{\mathcal{X}} \sigma}{\sqrt{2\pi}} \exp(-\frac{R^2}{2\sigma^2}) \geq \text{Cov}[X_i, h_\theta(X)], \quad \text{for all } i = 1, \ldots, d_X. \tag{34}$$

We can further bridge HSIC to adversarial robustness directly by taking advantage of the following lemma:

**Lemma 8** (Stein's Identity [15])**.** *Let $X = (X_1, X_2, \ldots X_{d_X})$ be multivariate normally distributed with arbitrary mean vector $\mu$ and covariance matrix $\Sigma$. For any function $h(x_1, \ldots, x_{d_X})$ such that $\frac{\partial h}{\partial x_i}$ exists almost everywhere and $\mathbb{E}|\frac{\partial}{\partial x_i}| < \infty$, $i = 1, \ldots, d_X$, we write $\nabla h(X) = (\frac{\partial h(X)}{\partial x_1}, \ldots, \frac{\partial h(X)}{\partial x_{d_X}})^\top$. Then the following identity is true:*

$$\mathrm{Cov}[X, h(X)] = \Sigma E[\nabla h(X)]. \tag{35}$$

*Specifically,*

$$\mathrm{Cov}\left[X_1, h\left(X_1, \ldots, X_{d_X}\right)\right] = \sum_{i=1}^{d_X} \mathrm{Cov}\left(X_1, X_i\right) E\left[\frac{\partial}{\partial x_i} h\left(X_1, \ldots, X_{d_X}\right)\right] \tag{36}$$

Given that $X \sim \mathcal{N}(0, \sigma^2 \mathbf{I})$, Lemma 8 implies:

$$\mathrm{Cov}\left[X_i, h_\theta\left(X\right)\right] = \sigma^2 \mathbb{E}\left[\frac{\partial}{\partial x_i} h_\theta\left(X\right)\right]. \tag{37}$$

Combining (34) and (37), we have:

$$\frac{RM_{\mathcal{Z}}}{M_{\mathcal{F}} M_{\mathcal{G}}} \mathrm{HSIC}(X, Z) + \frac{2M_{\mathcal{X}}\sigma}{\sqrt{2\pi}} \exp(-\frac{R^2}{2\sigma^2}) \geq \sigma^2 \mathbb{E}\left[\frac{\partial}{\partial x_k} h_\theta\left(X\right)\right]. \tag{38}$$

Note that a similar derivation could be repeated exactly by replacing $h_\theta(X)$ with $-h_\theta(X)$. Thus, for every $i = 1, 2, \ldots, d_X$, we have:

$$\frac{RM_{\mathcal{Z}}}{M_{\mathcal{F}} M_{\mathcal{G}}} \mathrm{HSIC}(X, Z) + \frac{2M_{\mathcal{X}}\sigma}{\sqrt{2\pi}} \exp(-\frac{R^2}{2\sigma^2}) \geq \sigma^2 \mathbb{E}\left[\left|\frac{\partial}{\partial x_i} h_\theta\left(X\right)\right|\right]. \tag{39}$$

Summing up both sides in (39) for $i = 1, 2, \ldots, d_X$, we have:

$$\frac{d_X R M_{\mathcal{Z}}}{M_{\mathcal{F}} M_{\mathcal{G}}} \mathrm{HSIC}(X, Z) + \frac{2d_X M_{\mathcal{X}}\sigma}{\sqrt{2\pi}} \exp(-\frac{R^2}{2\sigma^2}) \geq \sigma^2 \mathbb{E}\left[\sum_{i=1}^{d_X} \left|\frac{\partial}{\partial x_i} h_\theta\left(X\right)\right|\right]. \tag{40}$$

On the other hand, for $\delta \in \mathcal{S}_r$, by Taylor's theorem:

$$\mathbb{E}[|h_\theta(X + \delta) - h_\theta(X)|] \leq \mathbb{E}[|\delta^\top \nabla_X h_\theta(X)|] + o(r) \tag{41a}$$

$$\leq \mathbb{E}\left[\|\delta\|_\infty \|\nabla_X h_\theta(X)\|_1\right] + o(r) \tag{41b}$$

$$\leq r\mathbb{E}\left[\sum_{i=1}^{d_X} \left|\frac{\partial}{\partial x_i} h_\theta\left(X\right)\right|\right] + o(r), \tag{41c}$$

where (41b) is implied by Hölder's inequality, and (41c) is implied by the triangle inequality.

Combining (40) and (41), we have:

$$\frac{r d_X R M_{\mathcal{Z}}}{\sigma^2 M_{\mathcal{F}} M_{\mathcal{G}}} \mathrm{HSIC}(X, Z) + \frac{2r d_X M_{\mathcal{X}}}{\sqrt{2\pi}\sigma} \exp(-\frac{R^2}{2\sigma^2}) + o(r) \geq \mathbb{E}[|h_\theta(X + \delta) - h_\theta(X)|]. \tag{42}$$

Let $R = \sigma\sqrt{-2\log o(1)}$ where, here, $o(1)$ stands for an arbitrary function $w : \mathbb{R} \to \mathbb{R}$ s.t.

$$\lim_{r \to 0} w(r) = 0. \tag{43}$$

Then, we have $\frac{2r d_X M_{\mathcal{X}}}{\sqrt{2\pi}\sigma} \exp(-\frac{R^2}{2\sigma^2}) = o(r)$, because:

$$\lim_{r \to 0} \frac{2r d_X M_{\mathcal{X}}}{\sqrt{2\pi}\sigma} \exp(-\frac{R^2}{2\sigma^2})/r = \lim_{r \to 0} \frac{2d_X M_{\mathcal{X}}}{\sqrt{2\pi}\sigma} \exp(\log o(1))$$
$$= \lim_{r \to 0} \frac{2d_X M_{\mathcal{X}}}{\sqrt{2\pi}\sigma} o(1) \tag{44}$$
$$= 0$$

Thus, we conclude that:

$$\frac{r\sqrt{-2\log o(1)} d_X M_{\mathcal{Z}}}{\sigma M_{\mathcal{F}} M_{\mathcal{G}}} \mathrm{HSIC}(X, Z) + o(r) \geq \mathbb{E}[|h_\theta(X + \delta) - h_\theta(X)|]. \tag{45}$$

$\square$

## D  Licensing of Existing Assets

We provide the licensing information of each existing asset below:

**Datasets.**

- *MNIST mnist* is licensed under the Creative Commons Attribution-Share Alike 3.0 license.
- *CIFAR-10* and *CIFAR-100* [12] are licensed under the MIT license.

**Models.**

- The implementations of *LeNet* [17] and *ResNet-18* [11] in our paper are licensed under BSD 3-Clause License.
- The implementation of *WideResNet-28-10* [35] is licensed under the MIT license.

**Algorithms.**

- The implementations of *SWHB* [16], *PGD* [17], *TRADES* [36] are licensed under the MIT license.
- The implementation of *VIB* [1] is licensed under the Apache License 2.0.
- There are no licenses for *MART* [30] and *XIC* [9].

**Adversarial Attacks.** The implementations of *FGSM* [8], *PGD* [17], *CW* [3] and *AutoAttack* [5] are all licensed under the MIT license.

## E  Algorithm Details and Hyperparameter Tuning

Non-adversarial learning, information bottleneck based methds:

- *Cross-Entropy (CE)*, which includes only loss $\mathcal{L}$.
- *Stage-Wise HSIC Bottleneck (SWHB)* [16]: This is the original HSIC bottleneck. It does not include full backpropagation over the HSIC objective: early layers are fixed stage-wise, and gradients are computed only for the current layer.
- *XIC* [9]: To enhance generalization over distributional shifts, this penalty includes inputs and residuals (i.e., $\text{HSIC}(X, Y - h(X))$).
- *Variational Information Bottleneck (VIB)* [1]: this is a variational autoencoder that includes a mutual information bottleneck penalty.

Adversarial learning methods:

- *Projected Gradient Descent (PGD)* [17]: This optimizes $\mathcal{L}_r$, given by (3) via projected gradient ascent over $\mathcal{S}_r$.
- *TRADES* [36]: This uses a regularization term that minimizes the difference between the predictions of natural and adversarial examples to get a smooth decision boundary.
- *MART* [30]: Compared to TRADES, MART pays more attention to adversarial examples from misclassified natural examples and add a KL-divergence term between natural and adversarial examples to the binary cross-entropy loss.

We use code provided by authors, including the recommended hyperparameter settings and tuning strategies. In both SWHB and HBaR, we apply Gaussian kernels for $X$ and $Z$ and a linear kernel for $Y$. For Gaussian kernels, we set $\sigma = 5\sqrt{d}$, where $d$ is the dimension of the corresponding random variable.

We report all tuning parameters in Table 4. In particular, we report the parameter settings on the 4-layer LeNet [17] for MNIST, ResNet-18 [11] and WideResNet-28-10 [35] for CIFAR-10, and WideResNet-28-10 [35] for CIFAR-100 with the basic HBaR and when combining HBaR with state-of-the-art (i.e., PGD, TRADES, MART) adversarial learning.

For HBaR, to make a fair comparison with SWHB [16], we build our code, along with the implementation of PGD and PGD+HBaR, upon their framework. When combining HBaR with other state-of-the-art adversarial learning (i.e., TRADES and MART), we add our HBaR implemention to

the MART framework and use recommended hyperparameter settings/tuning strategies from MART and TRADES. To make a fair comparison, we use the same network architectures among all methods with the same random weight initialization and report last epoch results.

Table 4: Parameter Summary for MNIST, CIFAR-10, and CIFAR-100. $\lambda_x$ and $\lambda_y$ are balancing hyperparameters for HBaR; $\lambda$ is balancing hyperparameter for TRADES and MART.

| Dataset | param. | HBaR | PGD | PGD+HBaR | TRADES | TRADES+HBaR | MART | MART+HBaR |
|---|---|---|---|---|---|---|---|---|
| MNIST | $\lambda_x$ | 1 | - | 0.003 | - | 0.001 | - | 0.001 |
| | $\lambda_y$ | 50 | - | 0.001 | - | 0.005 | - | 0.005 |
| | $\lambda$ | | | - | 5 | 5 | 5 | 5 |
| | batch size | | 256 | | | 256 | | |
| | optimizer | | adam | | | sgd | | |
| | learning rate | | 0.0001 | | | 0.01 | | |
| | lr scheduler | | divided by 2 at the 65-th and 90-th epoch | | | divided by 10 at the 20-th and 40-th epoch | | |
| | # epochs | | 100 | | | 50 | | |
| CIFAR-10/100 | $\lambda_x$ | 0.006 | - | 0.0005 | - | 0.0001 | - | 0.0001 |
| | $\lambda_y$ | 0.05 | - | 0.005 | - | 0.0005 | - | 0.0005 |
| | $\lambda$ | | | - | 5 | 5 | 5 | 5 |
| | batch size | | 128 | | | 128 | | |
| | optimizer | | adam | | | sgd | | |
| | learning rate | | 0.01 | | | 0.01 | | |
| | lr scheduler | | cosine annealing | | | divided by 10 at the 75-th and 90-th epoch | | |
| | # epochs | 300 | 95 | | | 95 | | |

# F    Sensitivity of Regularization Hyperparameters $\lambda_x$ and $\lambda_y$

We provide a comprehensive ablation study on the sensitivity of $\lambda_x$ and $\lambda_y$ on MNIST and CIFAR-10 dataset with (Table 5 and 6) and without (Table 7 and 8) adversarial training. As a conclusion, (a) we set the weight of cross-entropy loss as 1, and empirically set $\lambda_x$ and $\lambda_y$ according to the performance on a validation set. (b) For MNIST with adversarial training, we empirically discover that $\lambda_x : \lambda_y$ ranging around $5 : 1$ provides better performance; for MNIST without adversarial training, $\lambda_x = 1$ and $\lambda_y = 50$, inspired by SWHB (Ma et al., 2020), provide the best performance. (c) for CIFAR-10 (and CIFAR-100), with and without adversarial training, $\lambda_x : \lambda_y$ ranging from $1 : 5$ to $1 : 10$ provides better performance.

Table 5: **MNIST by LeNet with adversarial training**: Ablation study on HBaR regularization hyperparameters $\lambda_x$ and $\lambda_y$ trained by HBaR +TRADES over the metric of natural test accuracy (%) and adversarial test robustness (PGD$^{40}$ and AA, %).

| $\lambda_x$ | $\lambda_y$ | Natural | PGD$^{40}$ | AA |
|---|---|---|---|---|
| 0.003 | 0.001 | 98.66 | 94.35 | 91.57 |
| 0.003 | 0 | 98.92 | 93.05 | 90.95 |
| 0 | 0.001 | 98.86 | 91.77 | 88.21 |
| 0.0025 | 0.0005 | 98.96 | 94.52 | 91.42 |
| 0.002 | 0.0005 | 98.92 | 94.13 | 91.33 |
| 0.0015 | 0.0005 | 98.93 | 94.06 | 91.43 |
| 0.001 | 0.0005 | 98.95 | 93.76 | 91.14 |
| 0.001 | 0.0002 | 98.92 | 94.61 | 91.37 |
| 0.0008 | 0.0002 | 98.94 | 94.15 | 91.07 |
| 0.0006 | 0.0002 | 98.91 | 94.13 | 90.72 |
| 0.0004 | 0.0002 | 98.90 | 93.96 | 90.56 |

Table 6: **CIFAR-10 by WideResNet-28-10 with adversarial training**: Ablation study on HBaR regularization hyperparameters $\lambda_x$ and $\lambda_y$ trained by HBaR +TRADES over the metric of natural test accuracy (%), and adversarial test robustness (PGD$^{20}$ and AA, %).

| $\lambda_x$ | $\lambda_y$ | Natural | PGD$^{20}$ | AA |
|---|---|---|---|---|
| 0.0001 | 0.0005 | 85.61 | 56.51 | 53.53 |
| 0.0001 | 0 | 80.19 | 49.49 | 45.33 |
| 0 | 0.0005 | 84.74 | 55.00 | 51.50 |
| 0.001 | 0.005 | 85.70 | 55.74 | 52.78 |
| 0.0005 | 0.005 | 84.42 | 55.95 | 52.66 |
| 0.00005 | 0.0005 | 85.37 | 56.43 | 53.40 |

Table 7: **MNIST by LeNet without adversarial training**: Ablation study on HBaR regularization hyperparameters $\lambda_x$ and $\lambda_y$ over the metric of HSIC$(X, Z_M)$, HSIC$(Y, Z_M)$, natural test accuracy (%), and adversarial test robustness (PGD$^{40}$, %).

| $\lambda_x$ | $\lambda_y$ | HSIC $(X, Z_M)$ | $(Y, Z_M)$ | Natural | PGD$^{40}$ |
|---|---|---|---|---|---|
| CE only | | 45.29 | 8.73 | 99.23 | 0.00 |
| 0.0001 | 0 | 21.71 | 8.01 | 99.28 | 0.00 |
| 0.001 | 0 | 5.82 | 6.57 | 99.36 | 0.00 |
| 0.01 | 0 | 3.22 | 4.28 | 99.13 | 0.00 |
| 0 | 1 | 56.45 | 9.00 | 98.92 | 0.00 |
| 0.001 | 0.05 | 53.70 | 8.99 | 99.13 | 0.03 |
| 0.001 | 0.01 | 10.44 | 8.51 | 99.37 | 0.00 |
| 0.001 | 0.005 | 8.86 | 8.24 | 99.38 | 0.00 |
| 0.01 | 0.5 | 16.13 | 8.90 | 99.14 | 5.00 |
| 0.1 | 5 | 15.81 | 8.90 | 98.96 | 7.72 |
| 1 | 50 | 15.68 | 8.89 | 98.90 | 8.33 |
| 1.1 | 55 | 15.90 | 8.88 | 98.88 | 6.99 |
| 1.2 | 60 | 15.76 | 8.89 | 98.95 | 7.24 |
| 1.5 | 75 | 15.62 | 8.89 | 98.94 | 8.23 |
| 2 | 100 | 15.41 | 8.89 | 98.91 | 7.00 |

Table 8: **CIFAR-10 by ResNet-18 without adversarial training**: Ablation study on HBaR regularization hyperparameters $\lambda_x$ and $\lambda_y$ over the metric of HSIC$(X, Z_M)$, HSIC$(Y, Z_M)$, natural test accuracy (%), and adversarial test robustness (PGD$^{20}$, %).

| $\lambda_x$ | $\lambda_y$ | HSIC $(X, Z_L)$ | $(Y, Z_L)$ | Natural | PGD$^{20}$ |
|---|---|---|---|---|---|
| CE only | | 3.45 | 4.76 | 95.32 | 8.57 |
| 0.001 | 0.05 | 43.48 | 8.93 | 95.36 | 2.91 |
| 0.002 | 0.05 | 43.15 | 8.92 | 95.55 | 2.29 |
| 0.003 | 0.05 | 41.95 | 8.90 | 95.51 | 3.98 |
| 0.004 | 0.05 | 30.12 | 8.77 | 95.45 | 5.23 |
| 0.005 | 0.05 | 11.56 | 8.45 | 95.44 | 23.73 |
| 0.006 | 0.05 | 6.07 | 8.30 | 95.35 | 34.85 |
| 0.007 | 0.05 | 4.81 | 8.24 | 95.13 | 15.80 |
| 0.008 | 0.05 | 4.44 | 8.21 | 95.13 | 8.43 |
| 0.009 | 0.05 | 3.96 | 8.14 | 94.70 | 10.83 |
| 0.01 | 0.05 | 4.09 | 7.87 | 92.33 | 2.90 |

Table 9: **MNIST by LeNet**: Mean and Standard deviation of natural test accuracy (in %) and adversarial robustness ((in %) on FGSM, PGD, CW, and AA attacked test examples) of adversarial learning baselines and combining HBaR with each correspondingly.

| Methods | MNIST by LeNet | | | | | |
|---|---|---|---|---|---|---|
| | Natural | FGSM | PGD$^{20}$ | PGD$^{40}$ | CW | AA |
| PGD | $98.40 \pm 0.018$ | $93.44 \pm 0.177$ | $94.56 \pm 0.079$ | $89.63 \pm 0.117$ | $91.20 \pm 0.097$ | $86.62 \pm 0.166$ |
| HBaR + PGD | $\mathbf{98.66} \pm 0.026$ | $\mathbf{96.02} \pm 0.161$ | $\mathbf{96.44} \pm 0.030$ | $\mathbf{94.35} \pm 0.130$ | $\mathbf{95.10} \pm 0.106$ | $\mathbf{91.57} \pm 0.123$ |
| TRADES | $\mathbf{97.64} \pm 0.017$ | $94.73 \pm 0.196$ | $95.05 \pm 0.006$ | $93.27 \pm 0.088$ | $93.05 \pm 0.025$ | $89.66 \pm 0.085$ |
| HBaR + TRADES | $\mathbf{97.64} \pm 0.030$ | $\mathbf{95.23} \pm 0.106$ | $\mathbf{95.17} \pm 0.023$ | $\mathbf{93.49} \pm 0.147$ | $\mathbf{93.47} \pm 0.089$ | $\mathbf{89.99} \pm 0.155$ |
| MART | $\mathbf{98.29} \pm 0.059$ | $95.57 \pm 0.113$ | $95.23 \pm 0.144$ | $93.55 \pm 0.018$ | $93.45 \pm 0.077$ | $88.36 \pm 0.179$ |
| HBaR + MART | $98.23 \pm 0.054$ | $\mathbf{96.09} \pm 0.074$ | $\mathbf{96.08} \pm 0.035$ | $\mathbf{94.64} \pm 0.125$ | $\mathbf{94.62} \pm 0.06$ | $\mathbf{89.99} \pm 0.13$ |

## G    Error Bar for Combining HBaR with Adversarial Examples

We show how HBaR can be used to improve robustness when used as a regularizer, as described in Section 4.2, along with state-of-the-art adversarial learning methods. We run each experiment by five times. Figure 5 illustrates mean and standard deviation of the natural test accuracy and adversarial robustness against various attacks on CIFAR-10 by ResNet-18 and WideResNet-28-10. Table 9, 10, 11, and 12 show the detailed standard deviation. Combined with the adversarial training baselines, HBaR consistently improves adversarial robustness against all types of attacks with small variance.

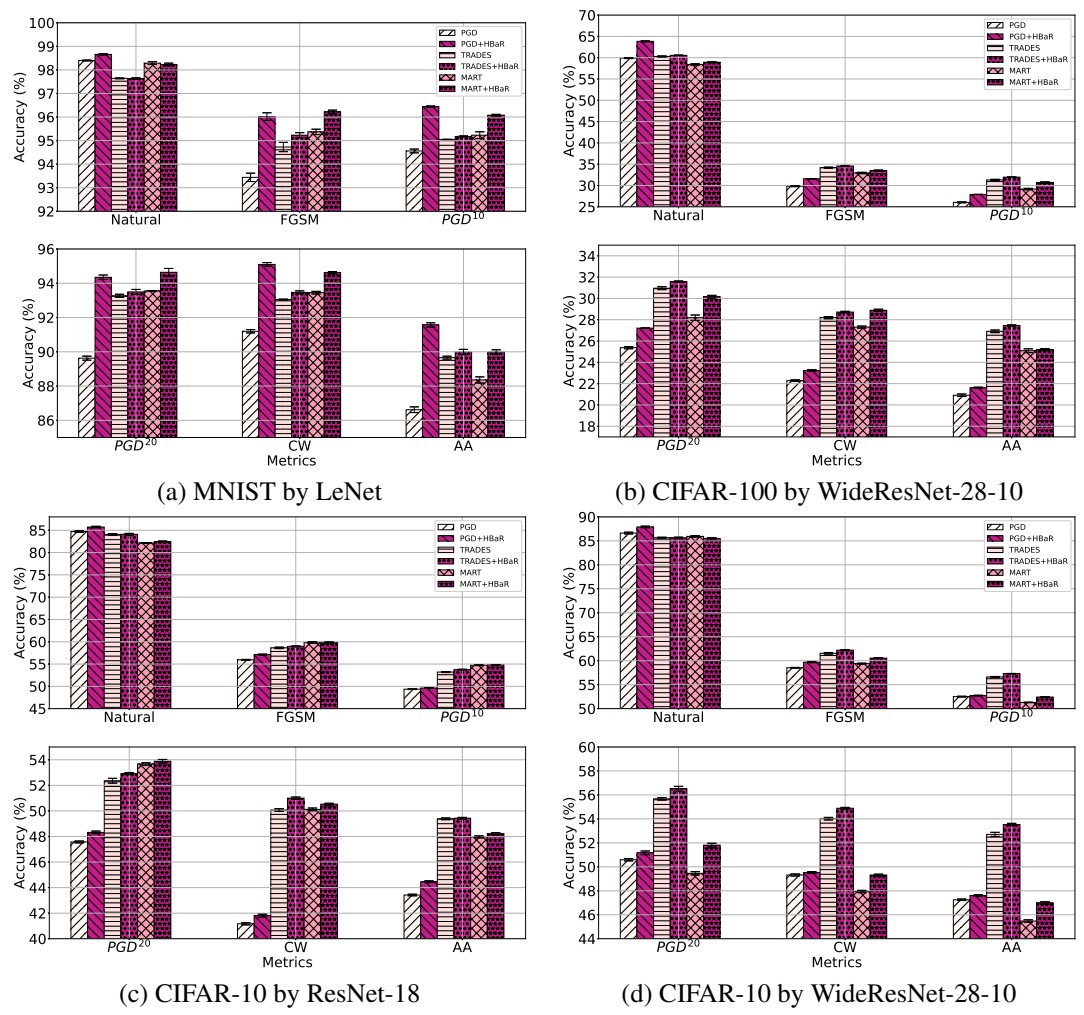

Figure 5: Error bar of natural test accuracy (in %) and adversarial robustness ((in %) on FGSM, PGD, CW, and AA attacked test examples) on MNIST by LeNet, CIFAR-100 by WideResNet-28-10, CIFAR-10 by ResNet-18 and WideResNet-28-10 of adversarial learning baselines and combining HBaR with each correspondingly.

Table 10: **CIFAR-10 by ResNet-18**: Mean and Standard deviation of natural test accuracy (in %) and adversarial robustness ((in %) on FGSM, PGD, CW, and AA attacked test examples) of adversarial learning baselines and combining HBaR with each correspondingly.

| Methods | CIFAR-10 by ResNet-18 | | | | | |
|---|---|---|---|---|---|---|
| | Natural | FGSM | $PGD^{10}$ | $PGD^{20}$ | CW | AA |
| PGD | 84.71±0.16 | 55.95±0.097 | 49.37±0.075 | 47.54±0.080 | 41.17±0.086 | 43.42±0.064 |
| HBaR + PGD | **85.73**±0.166 | **57.13**±0.099 | **49.63**±0.058 | **48.32**±0.103 | **41.80**±0.116 | **44.46**±0.169 |
| TRADES | 84.07±0.201 | 58.63±0.167 | 53.21±0.118 | 52.36±0.189 | 50.07±0.106 | 49.38±0.069 |
| HBaR + TRADES | **84.10**±0.104 | **58.97**±0.093 | **53.76**±0.080 | **52.92**±0.175 | **51.00**±0.085 | **49.43**±0.064 |
| MART | 82.15±0.117 | 59.85±0.154 | 54.75±0.089 | 53.67±0.088 | 50.12±0.106 | 47.97±0.156 |
| HBaR + MART | **82.44**±0.156 | **59.86**±0.132 | **54.84**±0.051 | **53.89**±0.135 | **50.53**±0.069 | **48.21**±0.100 |

Table 11: **CIFAR-10 by WideResNet-28-10**: Mean and Standard deviation of natural test accuracy (in %) and adversarial robustness ((in %) on FGSM, PGD, CW, and AA attacked test examples) of adversarial learning baselines and combining HBaR with each correspondingly.

| Methods | CIFAR-10 by WideResNet-28-10 | | | | | |
| --- | --- | --- | --- | --- | --- | --- |
| | Natural | FGSM | PGD$^{10}$ | PGD$^{20}$ | CW | AA |
| PGD | 86.63±0.186 | 58.53±0.073 | 52.21±0.084 | 50.59±0.096 | 49.32±0.089 | 47.25±0.124 |
| HBaR + PGD | **87.91**±0.102 | **59.69**±0.097 | **52.72**±0.081 | **51.17**±0.152 | **49.52**±0.174 | **47.60**±0.131 |
| TRADES | **85.66**±0.103 | 61.55±0.134 | 56.62±0.097 | 55.67±0.098 | 54.02±0.106 | 52.71±0.169 |
| HBaR + TRADES | 85.61±0.0133 | **62.20**±0.102 | **57.30**±0.059 | **56.51**±0.136 | **54.89**±0.098 | **53.53**±0.127 |
| MART | **85.94**±0.156 | 59.39±0.109 | 51.30±0.052 | 49.46±0.136 | 47.94±0.098 | 45.48±0.100 |
| HBaR + MART | 85.52±0.136 | **60.54**±0.071 | **53.42**±0.142 | **51.81**±0.177 | **49.32**±0.131 | **46.99**±0.137 |

Table 12: **CIFAR-100 by WideResNet-28-10**: Mean and Standard deviation of natural test accuracy (in %) and adversarial robustness ((in %) on FGSM, PGD, CW, and AA attacked test examples) of adversarial learning baselines and combining HBaR with each correspondingly.

| Methods | CIFAR-100 by WideResNet-28-10 | | | | | |
| --- | --- | --- | --- | --- | --- | --- |
| | Natural | FGSM | PGD$^{20}$ | PGD$^{40}$ | CW | AA |
| PGD | 59.91±0.116 | 29.85±0.117 | 26.05±0.106 | 25.38±0.129 | 22.28±0.079 | 20.91±0.133 |
| HBaR + PGD | **63.84**±0.105 | **31.59**±0.054 | **27.90**±0.030 | **27.21**±0.025 | **23.23**±0.088 | **21.61**±0.061 |
| TRADES | 60.29±0.122 | 34.19±0.132 | 31.32±0.134 | 30.96±0.135 | 28.20±0.097 | 26.91±0.172 |
| HBaR + TRADES | **60.55**±0.065 | **34.57**±0.068 | **31.96**±0.067 | **31.57**±0.079 | **28.72**±0.071 | **27.46**±0.098 |
| MART | 58.42±0.164 | 32.94±0.160 | 29.17±0.166 | 28.19±0.252 | 27.31±0.096 | 25.09±0.179 |
| HBaR + MART | **58.93**±0.102 | **33.49**±0.144 | **30.72**±0.130 | **30.16**±0.133 | **28.89**±0.118 | **25.21**±0.111 |

## H Limitations

One limitation of our method is that the robustness gain, though beating other IB-based methods, is modest when training with only natural examples. However, the potential of getting adversarial robustness *without* adversarial training is interesting and worth further exploration in the future. Another limitation of our method, as well as many proposed adversarial defense methods, is the uncertain performance to new attack methods. Although we have established concrete theories and conducted comprehensive experiments, there is no guarantee that our method is able to handle novel, well-designed attacks. Finally, in our theoretical analysis in Section 4.3, we have made several assumptions for Theorem 2. While Assumptions 1 and 2 hold in practice, the distribution of input feature is not guaranteed to be standard Gaussian. Although the empirical evaluation supports the correctness of the theorem, we admit that the claim is not general enough. We aim to proof a more general version of Theorem 2 in the future, hopefully agnostic to input distributions. We will keep track of the advances in the adversarial robustness field and further improve our work correspondingly.

## I Potential Societal Negative Impact

Although HBaR has great potential as a general strategy to enhance the robustness for various machine learning systems, we still need to be aware of the potential negative societal impacts it might result in. For example, over-confidence in the *adversarially-robust* models produced by HBaR as well as other defense methods may lead to overlooking their potential failure on newly-invented attack methods; this should be taken into account in safety-critical applications like healthcare [6] or security [26]. Another example is that, one might get insights from the theoretical analysis of our method to design stronger adversarial attacks. These attacks, if fall into the wrong hands, might cause severe societal problems. Thus, we encourage our machine learning community to further explore this field and be judicious to avoid misunderstanding or misusing of our method. Moreover, we propose to establish more reliable adversarial robustness checking routines for machine learning models deployed in safety-critical applications. For example, we should test these models with the latest adversarial attacks and make corresponding updates to them annually.