# OpenReview forum: "Revisiting Hilbert-Schmidt Information Bottleneck for Adversarial Robustness"
_NeurIPS.cc/2021/Conference — NeurIPS 2021 Poster_

### Official Review · Reviewer_cTjB · 2021-07-08

**Rating:** 7
**Confidence:** 4

**Summary:**

The authors propose to use an information bottleneck regularization -- with Hilbert-Schmidt information rather than mutual information -- to improve the adversarial robustness of the trained neural networks. The authors furthermore show that the minimizing HSIC between the input and the latent representation limits the variance the network output can achieve.

**Limitations And Societal Impact:**

Yes, the authors have partly addressed the societal impact by claiming that being too confident in adversarial robustness may be problematic. This being a method paper, I accept this statement as sufficient.

**Main Review:**

The main idea of the paper is very nice, especially in the light of the fact that for deterministic networks the classical IB objective is infinite (cf. [2] in the paper). The HSIC functionals appear not to be suffering from this limitation. However, I have noticed several small to medium issues that need to be addressed:

-) First of all, it is not exactly clear why the HSIC(Y;Z) term is required. Essentially, the classical or adversarial loss in (5) should suffice to ensure that Z contains sufficient information about Y, while HSIC(Y;Z) goes in the same direction. Moreover, in the classical IB setting, the term I(Y;Z) is usually replaced (not complemented) by the classical cross entropy loss. I have the strong feeling that it should be possible to remove this term altogether by appropriately adjusting the hyperparameters lambda_x. This is also supported by Table 3: In case HSIC(Y;Z) is missing during training, both HSIC terms turn out to be zero, indicating that the regularization on HSIC(X;Z) was too strong. This is further supported by Table 5, in which lambda_y=0 and for small lamda_x still good natural accuracy is achieved. Thus, HSIC(Y;Z) is not necessary, but a careful setting of lambda_x is required.

-) Theorem 1 is extremely strong and interesting; however, I agree with previous reviewers that it is not very useful for making the intended point. In fact, for a purely Gaussian setting, I expect that similar statements can be made about I(X;Z): If Z contains only little information about X (in the sense of a conditional variance, which appears to be the case for Gaussian settings and/our HSIC measured with distance-based kernels), then also the network output can only vary little with X. Theorem 2 goes in the same direction, essentially replacing variance with the first absolute moment under a specific perturbation model. Both theorems are interesting, but they do not explicitly point at adversarial robustness, other than just pointing at general invariance to input variations.

-) This brings me to a recent publication that may actually help relieving the previous issue. The conditional entropy bottleneck by Fischer replaces the I(X;Z) part in the IB functional by I(X;Z|Y). This term should indeed become zero, and setting this term to zero does not harm classification performance. If the same Theorems 1 and 2 could be formulated for HSIC(X;Z|Y) (appropriately defined), then this reduction in variation with the input would actually help make strong claims regarding generalization and adversarial robustness.

-) Some of the papers in the references are still referred to via their arXiv version, despite being already published. I suggest to update the reference list.

Aside from that, I noticed some minor issues that can be addressed quite easily. I list them here for completeness, they do not influence the review score though:
- [1] did not study learning dynamics, but [1] used the IB functional as a training objective. [2] did not suggest the IB functional for training, but analyzed it. [1] has a focus on supervised learning, not on auto-encoders. A reference to Fischer's conditional entropy bottleneck may be added.
- in line 92, while y in {0,1}^k is correct, this is not standard notation; y is usually one-hot encoded.
- after line 110, w.r.t. which distribution are the expectations taken?
- in lines 138-140, the computational complexity also depends on d_Z; this should be acknowledged.
- in line 171, why must the sets be closed? Is it not sufficient that they are bounded?
- in Table 1, the caption claims that training times are shown, but I think they are missing.
- line 244-245 contains a typo: "with and hyperparameters"
- line 273 lists Shwartz et al, but I guess it should be Shwartz-Ziv and Tishby.
- in Table 1 [ii] it is not surprising that missing the cross entropy loss yields low accuracy. Even if Z contains much information about Y, this information may not be easy to access for the remaining part of the network. This was discussed also in [2].


edit: I acknowledge the authors' responses and change my score. Thank you!

**Time Spent Reviewing:**

2

---

> ### Author Response · Authors · 2021-08-10
> **Response to Reviewer cTjB**
>
> We thank the reviewer for your detailed comments and constructive suggestions. We are also very encouraged by the positive feedback about the main idea of the paper. We have tried to address all your comments in the following:
>
> ### **Q1 Why HSIC(Y;Z) is Necessary**
> > First of all, it is not exactly clear why the HSIC(Y;Z) term is required. ...... Thus, HSIC(Y;Z) is not necessary, but a careful setting of lambda_x is required.
>
> We  had the same intuition as the reviewer initially, and believed that the second term HSIC(Y;Z) was redundant, in light of the presence of the cross entropy loss. Nevertheless, when we explored the effect of removing HSIC(Y; Z) we found that setting $\lambda_y = 0$ yields good natural accuracy, as the reviewer observed, but the same is not true for adversarial robustness. We observed that only by including both terms we can obtain adversarial robustness without adversarial training, while setting $\lambda_y = 0$ affects adversarial robustness significantly; this can also be seen in Table 5, rows 2-4.
> Please also note that the HSIC penalty involving Y is not equivalent to an output-only loss, like cross entropy. In particular, we add these penalty terms for every layer (see Eq. (5) and (6)), while the classical cross-entropy loss only directly affects representations on the output  layer. This may help explain the discrepancy between how the two of them impact adversarial accuracy.
>
> ### **Q2 Relate Theorems to Adversarial Robustness**
> >  Theorem 1 is extremely strong and interesting. ...... Both theorems are interesting, but they do not explicitly point at adversarial robustness, other than just pointing at general invariance to input variations.
>
> We agree with the reviewer that Theorem 1 is strong and interesting, as it bounds the variability of X; that is why we kept it in our current submission. We provided Theorem 2 to address exactly the concern raised by the reviewer (and previous reviewers): to connect this result to adversarial robustness (please note that the derivation of Theorem 2 was not a trivial extension). Allow us to point out that it is **not true** that we essentially replace variance with the first absolute moment *under a specific perturbation model*. The perturbations in $S_\delta$ are arbitrary, and the guarantee is *w.r.t. arbitrary such perturbations*. It is the samples X that are random and sampled from a Gaussian distribution. We use the latter assumption to make use of Stein’s lemma (linking variances to expectations of derivatives, and indirectly to sensitivity). The Gaussianity of inputs is by itself a restrictive technical assumption, but it is of course common.
>
> We also agree with the reviewer that Theorem 2 bounds robustness by showing that the model exhibits an invariance to input perturbations. We would like to point out however that this is very much consistent with Loss (3), various commonly encountered definitions of robustness (see, e.g., [18]), as well as intuition: a model is robust if a perturbing adversary cannot affect its output significantly. Finally, to highlight both the challenge in showing this as well as the nature of the contribution of both Theorems 1 and 2, these stability/robustness statements hold **for all** models. More specifically, they hold for all models for which the kernels are universal (see Assumptions 1 and 2), a condition which, as we discuss in the main text, holds if activation functions are continuous and bounded (e.g. softmax, sigmoid, etc.)
>
> ### **Q3 About Conditional Entropy Bottleneck**
> > This brings me to a recent publication that may actually help relieving the previous issue. ...... If the same Theorems 1 and 2 could be formulated for HSIC(X;Z|Y) (appropriately defined), then this reduction in variation with the input would actually help make strong claims regarding generalization and adversarial robustness.
>
> This is a very interesting direction, thank you for pointing this out to us. First, please note that our objective is already a proxy for this quantity. As Fischer notes, under the Markov chain assumptions he makes on X,Y, and Z: $$I(X;Y|Z)=I(X;Z)-I(Y;Z)$$
>
> (see Eq. (4) in Fischer). Moreover, his definition of the Conditional Entropy Bottleneck (CEB)” amounts to minimizing objective: $$I(X;Y|Z)- \gamma I(Y;Z)$$
>
> (see Eq. (5), in Fischer), which is of course equivalent to the Information Bottleneck (IB):
> $$I(X;Z)- \beta I(Y;Z)$$
>
> for $ \gamma= \beta-1$, as Fischer points out in Section 2.5.
> In other words, for appropriate choices of $\lambda_x$ and $\lambda_y$, our penalty becomes exactly what Fischer calls CEB, replacing mutual information with HSIC. Our work can thus be seen as an extension of both IB and CEB to a version based on HSIC, which is easier to compute  (and use as a penalty) compared to mutual information. This of course further motivates our choice of penalty and the use of both terms; we will cite the paper by Fischer and highlight these connections.
> Nevertheless, replacing this with a conditional HSIC would be an interesting direction. We can leverage the normalized conditional cross-covariance operator proposed by Fukumizu et al. to represent “HSIC(X; Z|Y)”. However, it is not trivial to establish concrete theorems for the new formulation as well as providing empirical evidence. Thus, we would like to explore this line of research as future work.
>
> ### **Q4 Update Reference List**
> > Some of the papers in the references are still referred to via their arXiv version, despite being already published. I suggest to update the reference list.
>
> We thank the reviewer for pointing this out. We will update our reference list in our final version.
>
> ### **Q5 Minor Issues**
> > [1] did not study learning dynamics, but [1] used the IB functional as a training objective. [2] did not suggest the IB functional for training, but analyzed it. [1] has a focus on supervised learning, not on auto-encoders. A reference to Fischer's conditional entropy bottleneck may be added.
>
> We will rephrase our descriptions for [1] and [2] more precisely according to your comment. Moreover, a reference to Fischer’s conditional entropy bottleneck will be added.
>
> > in line 92, while y in {0,1}^k is correct, this is not standard notation; y is usually one-hot encoded.
>
> We will update the notation of y to avoid any confusion.
>
> > after line 110, w.r.t. which distribution are the expectations taken?
>
> Sorry for the confusion. Please refer to Gretton et al. [10], Eq (8) for the exact notation of expectations, and we will update the formula accordingly in our final version.
>
> > in lines 138-140, the computational complexity also depends on d_Z; this should be acknowledged.
>
> We have assumed that calculating each kernel matrix entry takes constant time (line 138-139). However, we will acknowledge that the overall computational complexity also depends on $d_Z$ for clarity in our final version.
>
> > in line 171, why must the sets be closed? Is it not sufficient that they are bounded?
>
> Assumption 1 follows if the sets are closed and bounded (i.e., compact). The “closed” part is required for various reasons: e.g., to ensure continuity of $h_\theta$ and $g$ and that maxima in $C(X)$ and $C(Z)$ in Eq. (8) are attained.
>
> > in Table 1, the caption claims that training times are shown, but I think they are missing.
>
> We will update the caption of Table 1 and add a supplemental table containing training times in the appendix.
>
> > line 244-245 contains a typo: "with and hyperparameters"
>
> We will fix this typo in our final version.
>
> > line 273 lists Shwartz et al, but I guess it should be Shwartz-Ziv and Tishby.
>
> We will fix this inaccurate reference.
>
> > in Table 1 [ii] it is not surprising that missing the cross entropy loss yields low accuracy. Even if Z contains much information about Y, this information may not be easy to access for the remaining part of the network. This was discussed also in [2].
>
> We agree with this observation and we will add the reference to [2] when discussing the corresponding result.
>
> We thank the reviewer again for these valuable comments. Please let us know if you feel we haven’t fully addressed your comments. We will be happy to address them further.

---

> > ### Comment · Reviewer_cTjB · 2021-08-17
> > **Follow-Up**
> >
> > Thank you very much for the detailed answers. I agree with your points and I am willing to improve my score to eventually recommend acceptance. The answer to Q1 is satisfactory, the answer to Q2 clarifies many things, and the answer to Q3 -- that this is essentially out of scope -- is certainly valid. I will nevertheless follow up on Q2 and Q3, just to clarify some things:
> >
> > Ad Q2: I maintain the opinion that Th. 2 is strong, but I still question its usefulness. I.e., HSIC(X,Z) bounds how far an input perturbation can affect the network output. My point was that the whole point of training a neural network is that some kinds of input perturbations strongly affect the ouput -- namely those perturbations in X that are caused by changing Y. If the neural network would not react on changes in X induced by different classes/regression targets, then the network could not be used for classification/regression. I argue further that these "useful" perturbations may certainly be larger than the perturbations by an adversary. Thus, the bound in Th. 2 may be valid, but will be quite loose in practice since HSIC(X,Z) must be sufficiently large to allow classification/regression. (The same actually holds for I(X;Z).) Contrarily, HSIC(X,Z|Y) can be constrained to zero without affecting network performance for the intended task. This is why I believe that transferring the respective theorems to this conditional setup is more useful. (And I even think that this can be done trivially, i.e., by applying Th. 2 for HSIC(X,Z|Y=y) for every y individually.)
> >
> > Ad Q3: Indeed, it may be true that CEB and IB are equivalent in their information-theoretic formulations, but the same does not hold for the respective variational bounds. These variational bounds are the main difference between CEB and IB, since the chain rule does not apply to the variational bounds. Similarly, your line of argumentation only holds if you can show that HSIC -- the way it is computed -- satisfies a chain rule, such as mutual information does.

---

> > > ### Author Response · Authors · 2021-08-23
> > > **Follow-up Response to Reviewer cTjB**
> > >
> > > We thank the reviewer for raising the score and providing detailed clarifications on Q2 and Q3. We also would like to follow up with further discussions as these may lead to very interesting future works.
> > >
> > > ### Ad Q2
> > > > I maintain the opinion that Th. 2 is strong, ...... This is why I believe that transferring the respective theorems to this conditional setup is more useful. (And I even think that this can be done trivially, i.e., by applying Th. 2 for HSIC(X, Z|Y=y) for every y individually.)
> > >
> > > We agree with the reviewer that optimizing HSIC(X, Z) alone would ultimately suppress the classification/regression power of the neural network. That is also the motivation of including H(Y, Z) and the standard classification loss in addition to HSIC(X, Z).
> > > The additional penalties identify and reinforce directions that are discriminative. Theorem 2 is referring to only one of the terms we use in our objective, HSIC(X, Z), which deviates from the general "agree with the output" type of penalty. It is therefore not a full characterization of the entire method we employ, as captured by the full objective.
> > >
> > > We also agree with the reviewer that the conditional setup is potentially more useful. We note again the relationship between the combined objective HSIC(X, Z)-λHSIC(Y, Z) and what Fischer refers to as the "Conditional Entropy Bottleneck", as we noted above. This further highlights that our objective is taking the "directions that correlate with Y" into account, as the reviewer suggests, even if Theorem 2 (which again, captures only part of the penalty) does not.
> > >
> > > Extending the theorems to the conditional setup is definitely interesting. However, it may not be that trivial to do so. First, as we noted in our observation above, the right definition of HSIC(X, Z|Y) is hard to pinpoint; even doing so, deriving bounds is an additional challenge. Please also note that our approach does not directly generalize to terms of the form  HSIC(X, Z)-λHSIC(Y, Z), conditioned on Y=y or not: we derive an upper bound for HSIC(X, Z), while we would need a lower bound to incorporate HSIC(Y, Z), as the latter appears with a minus upfront in the penalty.
> > >
> > > Overall, this idea is very promising, and we thank the reviewer for suggesting it to us. We would very much like to further explore it as future work.
> > >
> > > ### Ad Q3
> > > > Indeed, it may be true that CEB and IB are equivalent in their information-theoretic formulations, but the same does not hold for the respective variational bounds. These variational bounds are the main difference between CEB and IB, since the chain rule does not apply to the variational bounds. Similarly, your line of argumentation only holds if you can show that HSIC -- the way it is computed -- satisfies a chain rule, such as mutual information does.
> > >
> > > We absolutely agree with the reviewer on this observation. When we define HSIC(X, Z|Y) properly, there may be proof showing that the chain rule somehow leads to a decoupled HSIC like the one we consider here. Nevertheless, there are many works that approximate/replace mutual information with HSIC ([1-3]). As long as the chain rule yields I(X, Z)-λΙ(Υ, Ζ), then HSIC could be used to approximate these two constituent terms respectively. Note that this is a heuristic, and does not correspond to bounds, but it is common, as in [4-7].
> > >
> > >
> > > [1] Song, Le, et al. "Feature Selection via Dependence Maximization." JMLR 2012.
> > >
> > > [2] Yokoi, Sho, et al. "Pointwise HSIC: A Linear-Time Kernelized Co-occurrence Norm for Sparse Linguistic Expressions." EMNLP 2018.
> > >
> > > [3] Wu, Chieh, et al. "Iterative spectral method for alternative clustering." AISTATS 2018.
> > >
> > > [4] Ma, Wan-Duo Kurt, et al. "The HSIC bottleneck: Deep learning without back-propagation." AAAI 2020.
> > >
> > > [5] Akhtaruzzaman, Md, et al. "HSIC bottleneck based distributed deep learning model for load forecasting in smart grid with a comprehensive survey." IEEE Access 2020.
> > >
> > > [6] Pogodin, Roman et al. “Kernelized information bottleneck leads to biologically plausible 3-factor Hebbian learning in deep networks.” NeurIPS 2020.
> > >
> > > [7] Li, Yazhe, et al. "Self-Supervised Learning with Kernel Dependence Maximization." arXiv preprint 2021.

---

> > > > ### Comment · Reviewer_cTjB · 2021-08-27
> > > > **Follow-up**
> > > >
> > > > Thanks a lot for engaging in the discussion, your points are certainly valid. I would appreciate continuing this discussion, but for now all my concerns have been addressed.

---

### Official Review · Reviewer_6H63 · 2021-07-16

**Rating:** 7
**Confidence:** 4

**Summary:**

To enhance the adversarial robustness of deep models, this paper proposes to use HSIC bottleneck as a regularizer. Unlike previous work trained with adversarial samples, the proposed learning objective drops redundant information from the input to the latent representations. Theoretical analysis that HSIC regularizer can reduce the output variance is provided.

**Ethics Review Area:**

["Privacy and Security (e.g., consent)"]

**Limitations And Societal Impact:**

Yes

**Main Review:**

This paper is valuable in the following aspects:
1.	The idea that improving robustness without adversarial samples is interesting. Although HSIC bottleneck is used in previous work, it is novel to use HSIC as a regular term for adversarial robustness.
2.	The theoretical analysis ensures the soundness that HSIC can reduce the influence of an adversarial attack.
3.	The effectiveness is well studied with ablations, and the experiment details are clear.
4.	The limitations and potential negative impacts are clearly discussed.

I only find some minor issues:
1.	It will be more readable if the abstract provides more information on the proposed method and theoretical conclusions.
2.	It is said that SWHB fails on ResNet-18 due to the way how SWHB updates parameters. However, Ma et al. have conducted experiments on ResNet and shows comparable performance with backprop. Could you please provide more explanation?
3.	Error bars in figure 5 are unclear. It might be better to show the standard deviations with a table.


**Needs Ethics Review:**

Yes

**Time Spent Reviewing:**

5

---

> ### Author Response · Authors · 2021-08-10
> **Response to Reviewer 6H63**
>
> We thank the reviewer for acknowledging the novelty of our idea, the soundness of our theory and the clearness of our empirical evaluations. We are very glad for the feedback and for highlighting the positive sides of our paper. Here, We have carefully answered all your questions in the following:
>
> ### **Q1 More Informative Abstract**
> > It will be more readable if the abstract provides more information on the proposed method and theoretical conclusions.
>
> We will update the abstract accordingly to make it more informative about our method and theoretical conclusions. The updated abstract reads as follows (the newly added sentences are shown in bold):
>
> “We investigate the HSIC (Hilbert-Schmidt independence criterion) bottleneck as a regularizer for learning an adversarially robust deep neural network classifier. **In addition to the usual cross-entropy loss, we add regularization terms for every intermediate output of the neural networks to ensure that the latent representations retain useful information for output prediction while reducing redundant information from the input.** We show that the HSIC bottleneck enhances robustness to adversarial attacks both theoretically and experimentally. **In particular, we prove that the HSIC bottleneck regularizer reduces the sensitivity of the classifier to adversarial examples.** Our experiments on multiple benchmark datasets and architectures demonstrate that incorporating an HSIC bottleneck regularizer attains competitive natural accuracy and improves adversarial robustness, both with and without adversarial examples during training. ”
>
> ### **Q2 Explain Why SWHB Fails**
> > It is said that SWHB fails on ResNet-18 due to the way how SWHB updates parameters. However, Ma et al. have conducted experiments on ResNet and shows comparable performance with backprop. Could you please provide more explanation?
>
> There are actually two variants of SWHB, dubbed “unformatted-training” and “format-training” by Ma et al. The result we showed in our paper is based on the “unformatted-trained” ResNet-18, while the result the reviewer refers to is based on the “format-trained” ResNet-18. We will add the result of the “format-trained” model for completeness. However, one can already observe in the original SWHB paper that the “format-trained” model could only achieve a natural accuracy less than 60%, which is far lower than our 95% natural accuracy.
>
> ### **Q3 Error Bar Table**
> > Error bars in figure 5 are unclear. It might be better to show the standard deviations with a table.
>
> We will add the table of standard deviations corresponding to the error bars in figure 5 in our final version.

---

> > ### Comment · Reviewer_6H63 · 2021-08-29
> > **Thank you for your response**
> >
> > Thank you for your response. All my concerns are addressed. I will keep my score of 7.

---

### Official Review · Reviewer_fUk4 · 2021-07-17

**Rating:** 7
**Confidence:** 4

**Summary:**

This paper proposes a novel regularization for adversarial robustness based on an information bottleneck criterion. The method is well-motivated and supported theoretically and empirically.

**Limitations And Societal Impact:**

Limitations and societal impact have been discussed in the appendix.

**Main Review:**

Strengths:
- The proposed regularization is well-motivated (Fig. 1, Lines 127-135).
- It is theoretically shown to upper bound the variance in the network output (Theorem 1 with continuity and bounded assumptions) and the expected absolute difference of the output (Theorem 2 with Gaussian input assumption). Theorem 2 is a more direct bound for the adversarial robustness.
- The experiments support the effectiveness of the idea in increasing the robustness of ResNet and WideResNet models on CIFAR-10 (Table 2) and CIFAR-100 (Table 1 right) when combined with other robustness methods as measured by various attacks such as AutoAttack. In particular, the improvements achieved combined with TRADES are significant (Table 1-right, CIFAR-100 ~0.5% against AA, Table 2-right, CIFAR-10 ~0.8% against AA). The improvements are shown to be statistically significant.

Minor:
- In Table 3 and Appendix E, the accuracy against PGD is low. Is this ablation study done without adversarial training? It would be better to see an ablation study on the best performing methods. Also, lambda_y is only set to 0.05 in Table 6 and a wider range needs to be studied.

**Time Spent Reviewing:**

2

---

> ### Author Response · Authors · 2021-08-10
> **Response to Reviewer fUk4**
>
> We sincerely thank the reviewer for these valuable comments. And we are encouraged to see your positive comments that this paper is a good work. We have tired to address your concern in the following response:
>
> ### **Q1 Ablation Study**
> > In Table 3 and Appendix E, the accuracy against PGD is low. Is this ablation study done without adversarial training? It would be better to see an ablation study on the best performing methods.
>
> The ablation study presented in Table 3 and Appendix E has been done without adversarial training. We agree that an ablation study on the best performing methods is a good complement to the whole story. However, due to limited response time, we can only conduct part of the ablation experiments by 8/10. In Table 1-ext-MNIST, which reports our ablation study on $\lambda_x$ and $\lambda_y$ for Table 1 using HBaR + PGD on MNIST dataset with adversarial training. We empirically discover that $\lambda_x : \lambda_y$ ranging from 2:1 to 5:1 provides better performance for MNIST dataset. The best performing one, i.e., $\lambda_x=0.003$,  $\lambda_y=0.001$, is what we reported in the paper. We will complete the experiments for CIFAR-10 dataset and include the results both here as well as it in our final version.
>
> Table 1-ext-MNIST: Ablation study on HBaR regularization hyperparameters \lambda_x and \lambda_y on MNIST by LeNet using HBaR + PGD with adversarial training, over the metric of natural test accuracy (%), and adversarial test robustness (PGD40, AA, %). The first line is what we reported in the paper.
>
> | $\lambda_x$      | $\lambda_y$    |      Natural         |  PGD40 |    AA        |
> |----------|------|---------|--|--------|
> |    **0.003**         |       **0.001**    |  **98.66**  |  **94.35**     | **91.57** |
> |    0.003               |       0         |     98.92          |   93.05 |   90.95   |
> |    0               |       0.001         |     98.86          |   91.77 |  88.21   |
> |    0.0025               |       0.0005         |     98.96  | 94.52   |    91.43   |
> |    0.002               |       0.0005         |     98.92    | 94.13      |    91.33   |
> |    0.0015             |       0.0005         |     98.93    | 94.06      |    91.43   |
> |    0.001               |       0.0005         |     98.95    | 93.76     |    91.14   |
> |    0.001               |       0.0002         |     98.92    | 94.61      |    91.37   |
> |    0.0008               |       0.0002        |     98.94   | 94.15       |    91.07   |
> |    0.0006             |       0.0002        |     98.91     | 94.13     |    90.72   |
> |    0.0004               |       0.0002         |     98.90  | 93.96     |    90.56   |
>
>
> ### **Q2 Range of $\lambda_y$**
> > Also, lambda_y is only set to 0.05 in Table 6 and a wider range needs to be studied.
>
> We agree that a more comprehensive range of hyperparameters should be presented. The ablation study in Table 6 focuses on the relative ratio between $\lambda_x$ and $\lambda_y$ without adversarial training. Thus, we fix $\lambda_y = 0.05$ and vary the ratio between $\lambda_x$ and $\lambda_y$ by only changing $\lambda_x$. In our experiments, we also observe that other ratios, as well as setting one of the coefficients to 0, result in almost zero adversarial robustness. We will include these empirical observations in our final version for completeness.

---

> > ### Comment · Reviewer_fUk4 · 2021-08-25
> > **Thank you for your response**
> >
> > I thank authors for their response. I keep my rating as accept. I encourage authors to incorporate their response into the paper.

---

### Review · Ethics_Reviewer_BPX7 · 2021-08-10

**Recommendation:**

The flagged issue is not a substantial ethical issue of the present research. The identified issue is minor and can be addressed in the discussion of the current version of paper. Specifically, the reviewer recommends acknowledging and addressing the possibility the adversarial robustness might not be desirable, or might be harmful, in connection to the presented method. This can be done the discussion of potential negative societal impacts in the Conclusions or Appendix H.

**Ethics Review:**

The paper describes a regularizer that improves adversarial robustness. The method should enable building more adversarially robust ML models, precluding external manipulations.

The paper does a sufficient job at acknowledging security harms and the risks of over-confidence in the discussion of negative societal impacts.

However, whereas in many deployments of ML making a model adversarially robust is a good thing, in some it is not. Adversarial robustness enables to make harmful systems more resilient to both technologically advanced evasion attempts, as well as to ad-hoc evasion attempts. In image domains, this means, e.g., fewer ways to ensure privacy against privacy-invasive facial recognition systems. Outside of image domains, adversarial robustness is at conflict with desirable properties such as enabling actionable recourse [1]. See, e.g., the work of Albert, Penny, Schneier, Shankar [2] for the detailed discussion of such negative societal effects in adversarial machine learning.

[1] Actionable Recourse in Linear Classification. Berk Ustun, Alexander Spangher, Yang Liu. 2018
[2] Politics of Adversarial Machine Learning. Kendra Albert, Jonathon Penney, Bruce Schneier, Ram Shankar Siva Kumar. 2020.

---

### Decision · Program_Chairs · 2021-09-27

**Decision:**

Accept (Poster)

**Comment:**

Adversarial robustness is an important problem of neural network models. This paper proposed a new regularization based on Hilbert-Schmidt Independence Criterion (HSIC) to improve adversarial robustness. The regularization consists two HSIC terms: one aims to reduce the nonlinear dependence between input and features, while the other enhances the dependence between features and outputs. Such a regularization is shown to enhance adversarial robustness in both natural training and adversarial training. The reviewers unanimously accept this paper after discussions. There are several minor points raised by reviewers that the authors promise to revise accordingly. In the preparation of final version, the authors are expected to incorporate these points suggested by the reviewers.